# Targeting N-myristoylation for therapy of B-cell lymphomas

Erwan Beauchamp[1,2], Megan C. Yap[1,2], Aishwarya Iyer[1], Maneka A. Perinpanayagam[1,2], Jay M. Gamma[3], Krista M. Vincent[4], Manikandan Lakshmanan[5], Anandhkumar Raju[6,7], Vinay Tergaonkar[6,7], Soo Yong Tan[6,7], Soon Thye Lim [8], Wei-Feng Dong[4], Lynne M. Postovit [4], Kevin D. Read [9], David W. Gray [9], Paul G. Wyatt[9], John R. Mackey[2,4] & Luc G. Berthiaume [1,2✉]

Myristoylation, the N-terminal modification of proteins with the fatty acid myristate, is critical for membrane targeting and cell signaling. Because cancer cells often have increased N-myristoyltransferase (NMT) expression, NMTs were proposed as anti-cancer targets. To systematically investigate this, we performed robotic cancer cell line screens and discovered a marked sensitivity of hematological cancer cell lines, including B-cell lymphomas, to the potent pan-NMT inhibitor PCLX-001. PCLX-001 treatment impacts the global myristoylation of lymphoma cell proteins and inhibits early B-cell receptor (BCR) signaling events critical for survival. In addition to abrogating myristoylation of Src family kinases, PCLX-001 also promotes their degradation and, unexpectedly, that of numerous non-myristoylated BCR effectors including c-Myc, NFκB and P-ERK, leading to cancer cell death in vitro and in xenograft models. Because some treated lymphoma patients experience relapse and die, targeting B-cell lymphomas with a NMT inhibitor potentially provides an additional much needed treatment option for lymphoma.

[1] Department of Cell Biology, Faculty of Medicine and Dentistry, University of Alberta, Edmonton T6G 2H7 AB, Canada. [2] Pacylex Pharmaceuticals Inc., Edmonton, AB, Canada. [3] Departments of Medicine, Faculty of Medicine and Dentistry, University of Alberta, Edmonton T6G 2H7 AB, Canada. [4] Departments of Oncology, Faculty of Medicine and Dentistry, University of Alberta, Edmonton T6G 2H7 AB, Canada. [5] Mouse Models of Human Cancer Unit, Institute of Molecular and Cell Biology, 61 Biopolis Drive, Proteos 138673, Singapore. [6] Advanced Molecular Pathology Lab, Institute of Molecular and Cell Biology, 61 Biopolis Drive, Proteos, Singapore 138673. [7] Department of Pathology, National University of Singapore, Singapore, Singapore. [8] Department of Medical Oncology, National Cancer Centre Singapore, 11 Hospital Drive, Outram Road, Singapore 169610, Singapore. [9] Drug Discovery Unit, School of Life Sciences, University of Dundee, James Black Centre, Dow Street, Dundee DD1 5EH, UK. ✉email: luc.berthiaume@ualberta.ca

Hematological cancers such as lymphoma account for ~9% of new cancer cases and cancer-related deaths worldwide[1–3]. Although patients with aggressive non-Hodgkin lymphomas such as Burkitt lymphoma (BL) and diffuse large B-cell lymphoma (DLBCL) frequently achieve initial remission with current therapies, these are toxic and a substantial proportion of patients experience disease relapse and premature death[2,3]. Recent data from the Surveillance, Epidemiology, and End Results of the National Cancer Institute (NCI) show a 5-year post diagnosis survival rate for non-Hodgkin lymphoma and DLBCL, relative to age-matched controls, of only 70% and 63%, respectively[2]. The identification of new druggable targets and better-tolerated treatments for aggressive lymphomas are therefore much needed.

While B-cell receptor (BCR) signaling is essential for normal B-cell function, it is often deregulated and provides critical pro-survival signals for B-cell lymphomagenesis in both BL and DLBCL[4–8]. Indeed, the presence of self-antigens and/or mutations in key BCR effectors impact distinct signaling modes of the BCR. In addition to the ligand activated BCR signaling mode, these include the chronic active BCR signaling in activated B cell-like DLBCL and chronic lymphocytic leukemia as well as the tonic (antigen independent constitutive baseline signaling) BCR signaling in BLs[4–8]. Typically, engagement of the BCR leads to the translocation of this receptor to plasma membrane lipid rafts containing the myristoylated Src-family kinase (SFK) Lyn[9–11]. Myristoylated Lyn phosphorylates select tyrosine residues in the immune-receptor tyrosine-based motif (ITAM) of the BCR associated CD79A-CD79B heterodimer[12,13] resulting in the recruitment of spleen tyrosine kinase (SYK). Human germinal center-associated (HGAL) protein is another myristoylated protein localized to lipid rafts and is phosphorylated upon BCR activation[14,15]. Phosphorylated HGAL enhances BCR signaling by augmenting the activation and recruitment of SYK to phosphorylated ITAMs, triggering the tyrosine phosphorylation of the Tec family member Bruton's tyrosine kinase (BTK)[16], phospholipase $C_\Upsilon$, and protein kinase $C\beta$[13]. Activated phospholipase $C_\Upsilon$ activity produces diacylglycerol and inositol-trisphosphate ($IP_3$), which activate PKCs and mobilize calcium ions from endoplasmic reticulum stores respectively. These chemical mediators, in turn, activate various signaling pathways[17]. All these early signaling events promote cell survival and proliferation through activation of transcription via the NFκB, PI3K, extracellular signal regulated kinase (ERK) mitogen-activated protein kinase, CREB, and NF-AT pathways[4–6,18]. The importance of BCR signaling in lymphomagenesis has prompted the development of numerous pharmacological agents, which target effector proteins downstream of the BCR including various SFKs (dasatinib), BTK (ibrutinib), and PI3Kδ (CAL-101)[4,5,19,20].

In humans, protein myristoylation is mediated by two ubiquitously expressed N-myristoyl-transferases, NMT1, and NMT2, which add a 14 carbon fatty acid myristate onto numerous proteins[21,22]. Myristoylation plays a fundamental role in cell signaling and allows for the dynamic interactions of proteins with cell membranes[23,24]. Myristoylation occurs at the N-terminal glycine residue of proteins either co-translationally after the removal of the initiator methionine or post-translationally after caspase-cleavage during apoptosis[23]. Up to 600 proteoforms[25] in humans are myristoylated and the proper membrane targeting and functions of these proteins require myristoylation[23,24,26–28]. SFKs, Abl, $G_\alpha$ subunits, Arf GTPases, caspase truncated (ct-) Bid, and ct-PAK2 are examples of myristoylated proteins that critically regulate cell growth and apoptosis[23,29–35]. Recently, NMTs were also shown to be responsible for myristoylation of N-terminally located lysine residues of Arf6 GTPase, thereby adding to their roles in cell signaling[36,37]. Because NMTs are essential for the viability of parasites, small molecule inhibitors such as DDD85646 were developed as a *T. brucei* NMT inhibitor to treat African sleeping sickness[38]. DDD85646 was also synthesized and validated independently as a bona fide inhibitor of human NMTs under the name IMP-366[39]. Because NMT expression levels and activity are increased in some cancers[40–45], NMTs have been proposed to be anticancer targets[43]. However, the effect of NMT inhibitors in cancer has not been systematically investigated.

Herein, we tested the sensitivity of 300 cancer cell lines encompassing all major cancer types to NMT inhibition by PCLX-001 in three independent screens. PCLX-001 is an orally bioavailable derivative of the NMT inhibitor DDD85646, and is more selective and potent towards human NMTs (Supplementary Table S1)[38]. We demonstrate that PCLX-001 inhibits the viability and growth of hematological cancer cells in vitro more effectively than the inhibition of viability and growth of other cancer cell types or select normal cells. PCLX-001 disrupts early BCR-mediated survival signaling in several B-cell lymphoma cell lines and promotes the degradation of numerous myristoylated and non-myristoylated BCR effectors, triggering apoptosis. More importantly, PCLX-001 produces dose-dependent tumor regression and complete tumor regressions in two of three lymphoma murine xenograft models establishing an initial proof-of-concept for NMT inhibitors as cancer therapeutics and supporting its ongoing preclinical development.

## Results

**PCLX-001 selectively kills blood cancer cells in vitro.** To investigate the therapeutic potential of NMT inhibition in cancer, we performed three independent robotic screens to measure the percentage growth inhibition (GI) of PCLX-001 in a variety of cancer cell lines. Using 68 cell lines on the Horizon (St. Louis, MO) platform, we show PCLX-001 inhibits the growth of a variety of cell lines (Fig. 1a). GI is significantly higher ($P < 0.0001$) however, in hematological (blood) cancer cells including lymphomas, leukemia, and myelomas than in other cancer cell line types (Fig. 1b). These results were recapitulated using a 101 cell line Oncolines[TM] (Oss, Netherlands) screen (Fig. 1c, d, $P = 0.0001$, Supplementary Fig. 1), and in a third screen (Chempartner, Shanghai, China) whereby 131 cancer cell lines were exposed to PCLX-001 for 3 and 6 days (Supplementary Fig. 2). The median $IC_{50}$ following 3 days of PCLX-001 treatment is significantly lower in hematological cancer cell lines (0.166 μM) in comparison to cell lines originating from solid tumors (10 μM, the highest dose tested; $P = 0.0038$) including breast cancer, non-small cell lung cancer, and small cell lung cancer (Supplementary Fig S2A, B). By day 6 however, PCLX-001 effectively kills nearly all types of cancer cell lines tested (Supplementary Fig S2C, D).

To confirm the data obtained using screens, we tested the effects of PCLX-001 treatment on several common B-cell lymphoma cell lines including the BL cell lines BL2, Ramos, and BJAB, the DLBCL cell lines DOHH2, WSU-DLCL2, and SU-DHL-10, and the immortalized B-cells IM9 and VDS[46] as controls. We performed three types of assays on these cells: (1) CellTiter Blue assay, whose readouts are dependent on both proliferation (number of total cells) and viability (percentage of viable cells) to evaluate the total number of viable cells, (2) Calcein assay, whose readout is independent of proliferation rate as it only measures the percentage of viable cells and, (3) a cell proliferation assay to simply count the total number of cells over time, independently of their viability. Incubation of malignant cell lines with PCLX-001 kills these cells in a time and concentration dependent manner in all three assays. Furthermore, PCLX-001 treatment kills malignant cell lines at significantly lower concentrations than that needed to kill benign IM9 and VDS

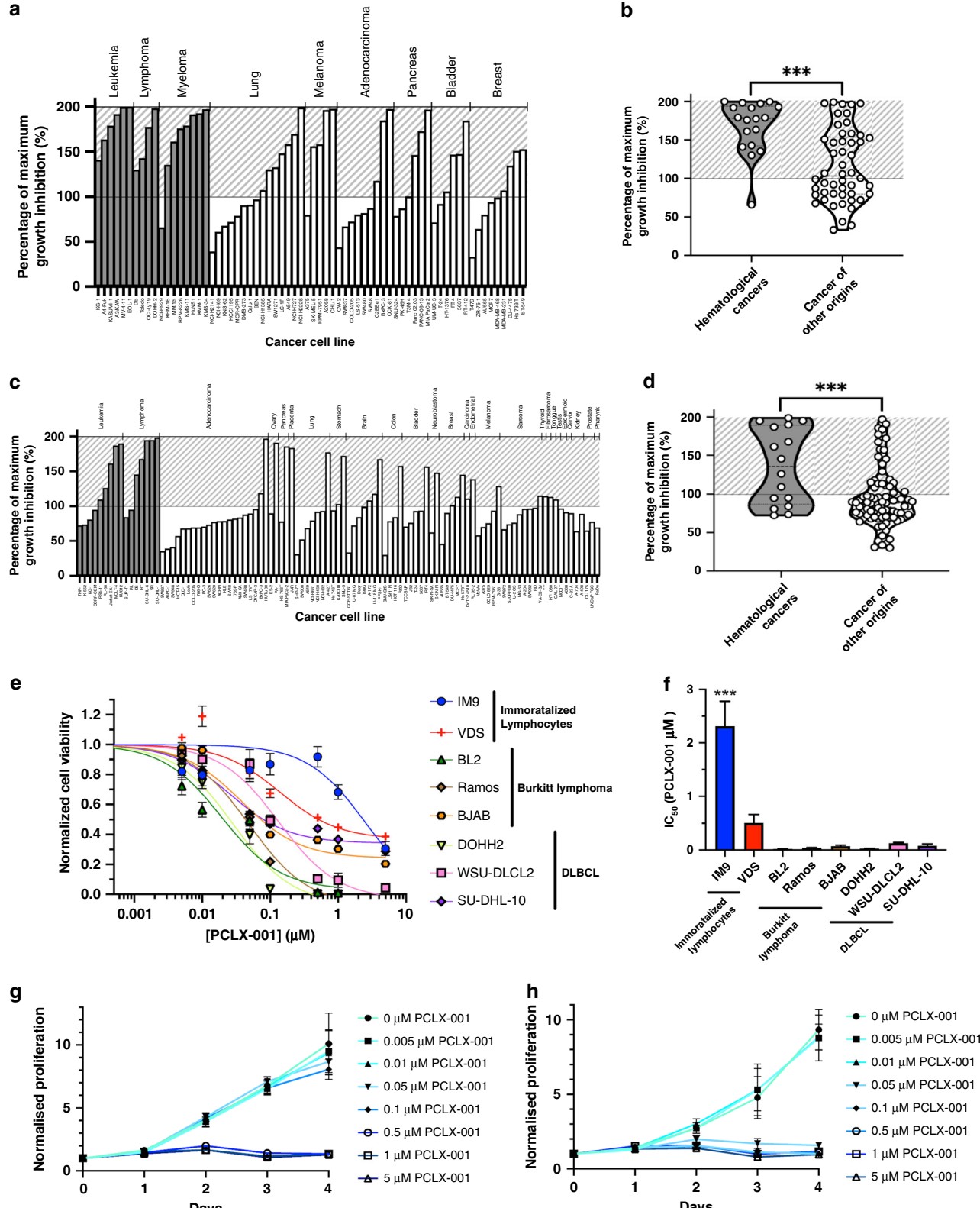

B-cells as measured by both CellTiter Blue (4 to 111 fold less PCLX-001 needed; Fig. 1e, f) and Calcein (2 to 40 fold less PCLX-001 needed; Supplementary Fig. 3) assays. PCLX-001 is also better at inhibiting the proliferation and viability of the six malignant B-lymphoma cell lines in comparison to benign IM9 and VDS B-cells (Fig. 1e–h, Supplementary Figs. S3 and S4). To illustrate this, we show the treatment of malignant BL2 cells with 0.05 and

0.1 μM PCLX-001 completely inhibits their proliferation over time with little effect on benign IM9 cells at these concentrations (Fig. 1g, h). Importantly, 96 h PCLX-001 treatment of freshly isolated human lymphocytes and peripheral blood mononuclear cells (PBMCs) only marginally affects lymphocyte survival, whereas 0.1 μM PCLX-001 causes an ~50% decrease in PBMC survival (Supplementary Fig. 5). The surviving PBMCs however,

**Fig. 1 PCLX-001 selectively kills hematological cancer cell lines in comparison to cancer cell lines of other origins.** Percentage of maximum growth inhibition of various cell lines following 96 h treatment with 1.2 μM PCLX-001 as determined using a Horizon cell line screen (**a**, **b**), or following 72 h treatment with 1 μM of PCLX-001 using a OncolinesTM cell line screen (**c**, **d**). Cell lines are arranged according to tumor cell type. Cross-hatched zone represents cytotoxic effect. Hematological cancer cell lines are depicted in gray while all other types of cancer cell lines are depicted in white. Corresponding violin graphs compare the average PCLX-001-mediated growth inhibition on hematological cancer cell lines to cancer cell lines of other origins combined as calculated from the Horizon (**b**) and OncolinesTM (**d**) cell screens (Unpaired $t$ test, two-tailed $P < 0.0001$). Quartiles are separated by dotted lines. Error bars represent standard deviation within each group. Normalized cell viability curves of immortalized lymphocyte (IM9, VDS), BL (BL2, Ramos, BJAB), and DLBCL (DOHH2, WSU-DLCL2, SU-DHL-10) cell lines treated with 0.001–5 μM of PCLX-001 for 96 h as determined by CellTiter Blue Viability Assay (**e**). Corresponding histograms of absolute $IC_{50}$ (and SD) values calculated from a log(inhibitor) vs response (three parameters) equation cell viability curves plotted in **e** (**f**). ***Indicates a significant difference ($P \leq 0.001$) in IC50 between IM9 cells and all other cell lines tested (Ordinary one-way Anova, Tukey's multiple comparisons test, $P < 0.0001$). Normalized proliferation of IM9 (**g**) and BL2 (**h**) cells reated with 0–5 μM of PCLX-001 for 96 h as determined by cell count. Values are mean ± s.e.m. of three independent experiments. Source data are provided as a Source Data file.

endure PCLX-001 treatment up to a concentration of 10 μM, a dose ~100× greater than the $IC_{50}$ (~0.050–0.100 μM) for most hematological cancer cell lines in vitro. A similar trend is observed in primary human umbilical vein endothelial cells (HUVECs; Supplementary Fig. 5B). Taken together, these data show that PCLX-001 treatment selectively inhibits the proliferation and viability of a variety of cancer cell lines in a time and concentration dependent manner, and is particularly efficient at killing malignant hematologic cancer cells in vitro.

**Myristoylation inhibition induces lymphoma cell apoptosis.** To verify that PCLX-001 acts on target, we used click chemistry as described[47] to visualize the inhibition of endogenous protein myristoylation in malignant BL2 lymphoma cells and benign IM9 B-cells (Fig. 2a, b). PCLX-001 inhibits total protein myristoylation in a concentration dependent manner in both cell lines. However, only ~0.1 μM of PCLX-001 is required to decrease BL2 myristoylation compared to five times this amount in IM9 cells (Fig. 2a, b). This suggests that protein myristoylation processes in malignant BL2 cells may somehow be more sensitive to PCLX-001 inhibition. Although PCLX-001 (Supplementary Table S1)[38] is a closely related analog of DDD85646/IMP-366 and part of a series of recently validated NMT inhibitors[38,39], we further evaluated its effect on palmitoylation and phosphorylation. PCLX-001 does not inhibit the palmitoylation of an EGFP-N-Ras construct expressed in COS-7 cells (Supplementary Fig. 6A), nor does it significantly inhibit any of the 468 human kinases of the pre-configured scanMAX KINOMEscan™ (Eurofins DiscoverX, San Diego, USA) at concentrations up to 10 μM (Supplementary Fig. 6B). Of note, only three possible positive hits were found at 100 μM PCLX-001, a concentration ~4000× greater than the EC50 of PCLX-001 for BL2 cells. Thus, the time and concentration dependent effects of PCLX-001 on cellular function and viability appear NMT-specific.

We next verified PCLX-001 inhibition of NMT function by monitoring the myristoylation and localization of Src protein tyrosine kinase, a known myristoylated protein, using truncated Src-EGFP[48] constructs expressed in COS-7 cells by click chemistry[47] and fluorescence microscopy. PCLX-001 inhibits the myristoylation of both the WT-Src-EGFP construct and endogenous Src in a concentration dependent manner in COS-7 and IM9 cells, respectively (Fig. 2c, d). Notably, myristoylation inhibition relocalizes WT-Src-EGFP from the plasma and endosomal membranes to the cytoplasm in COS-7 cells, producing a distribution pattern comparable to that of the non-myristoylatable Gly2Ala-Src-EGFP mutant construct[48] (Fig. 2e). Inhibiting endogenous Src myristoylation also produces an unexpected time-dependent reduction in Src protein levels in BL2 and IM9 cells treated with PCLX-001 for up to 5 days (Fig. 2f) that is accelerated in malignant BL2 cells in comparison

to IM9 controls ($P = 0.0174$; Supplementary Fig. S7). Furthermore, PCLX-001 treatment selectively induces apoptosis in the BL cell lines BL2 and Ramos, but not immortalized IM9 B-cells as measured by PARP-1 and caspase-3 cleavage (Fig. 2g), consistent with benign, immortalized B-cells exhibiting a higher threshold for PCLX-001 toxicity (Fig. 1e, f, Supplementary Fig. 3). Altogether, these data suggest that PCLX-001 preferentially abrogates myristoylation in malignant lymphoma cells in comparison to normal immortalized B cells leading to selective cell death.

**PCLX-001 reduces SFK levels and BCR downstream signaling.** BCR signaling provides key survival signals in B-cell lymphomas, and SFKs (especially Lyn) play a critical role in initiating BCR signaling in both normal B-cells and lymphomas[5,6,11,49,50]. Since PCLX-001 treatment preferentially reduces endogenous Src protein levels in malignant BL2 cells in comparison to benign IM9 controls (Fig. 2f), we sought to determine if a similar effect could be observed on other SFKs in various lymphoma cell lines. We found that PCLX-001 treated BL2, Ramos, BJAB, DOHH2, WSU-DLCL2, and SU-DHL-10 lymphoma cells all exhibit a more pronounced dose and time dependent decrease in Src and Lyn SFK protein levels in comparison to benign IM9 and VDS controls (Fig. 3a). To investigate whether the proteasome degradation mechanism was involved, after addition of PCLX-001 to BL2 cells for 24 or 48 h, we treated BL2 cells with the proteasome inhibitor MG132 or not for 6 h prior to harvesting and lysing the cells, and, measuring residual protein levels of not only Src and Lyn SFKs, but also hematopoietic cell kinase (Hck) and lymphocyte specific kinase (Lck) SFKs, both of which are also linked to lymphoma progression[50]. PCLX-001 treatment reduces Hck and Lck protein levels to a lesser degree than Src and Lyn (Fig. 3b). However, the addition of MG132 to PCLX-001 treated cells results in partial or complete restoration of the 4 SFK proteins in comparison to controls, especially at the 24h time point (Fig. 3b). This indicates that the degradation of non-myristoylated-SFKs can be attributed in part to the ubiquitin-proteasome system. The efficacy of the proteasome inhibition by MG132 was confirmed by monitoring Mcl-1 levels, a protein actively degraded by the proteasome[51] (Fig. 3b).

Because antigen independent basal BCR signaling is often elevated in lymphoma cells[6,49], we assessed the impact of PCLX-001 treatment on ligand independent BCR signaling by monitoring endogenous tyrosine phosphorylation levels in the above cell lines using an anti-phospho-tyrosine (P-Tyr) antibody (PY99). 24 h treatments with PCLX-001 decreases antigen independent global phospho-tyrosine levels in all cell lines tested in a concentration dependent manner (Supplementary Fig. 8A). In addition, 1 μM PCLX-001 abrogates nearly all ligand dependent BCR mediated phospho-tyrosine and pan-phospho-SFK levels in BL2 cells after BCR ligation with anti-IgM (Fig. 3b). While

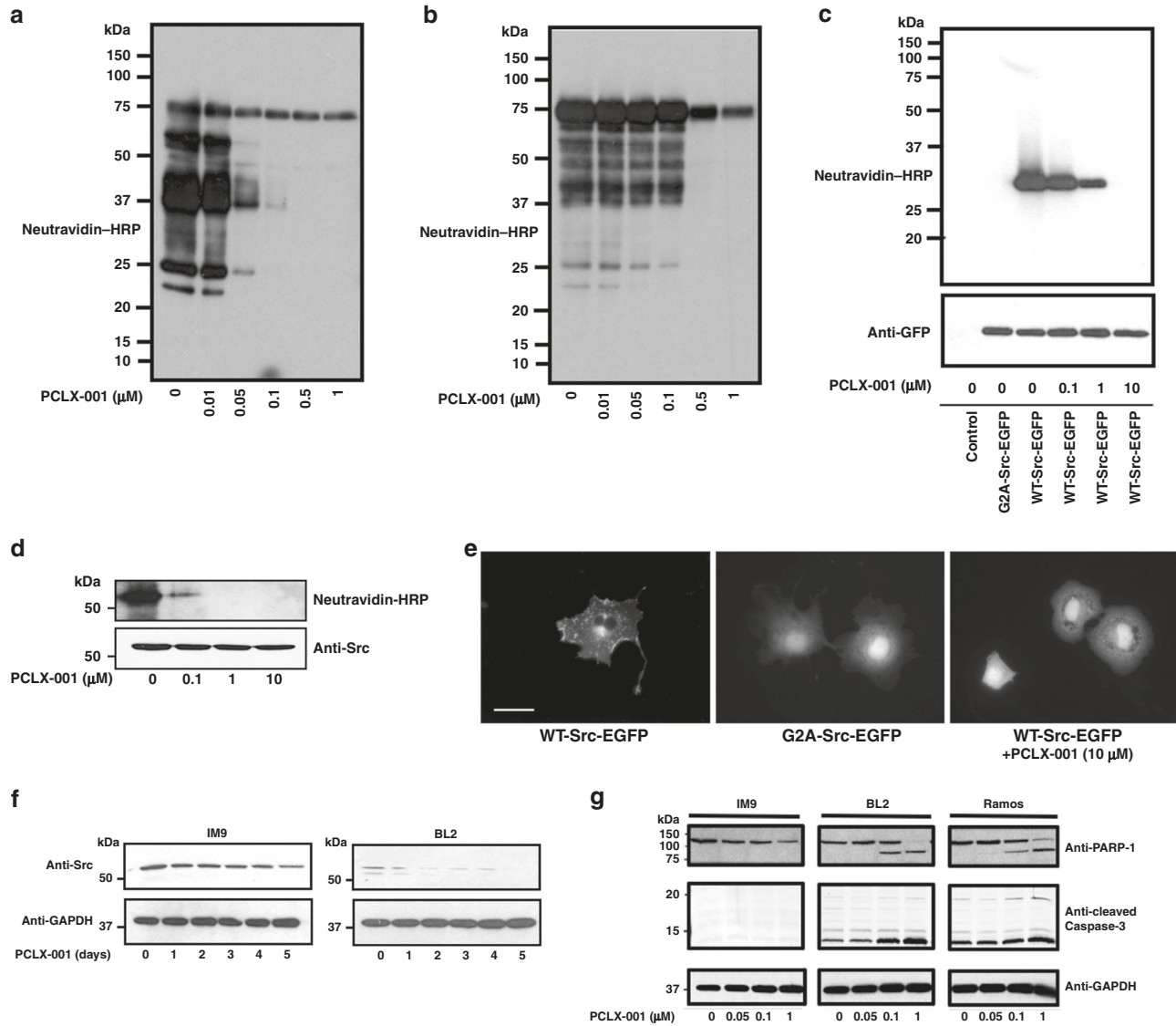

**Fig. 2 PCLX-001 selectively inhibits myristoylation in vitro and induces apoptosis in lymphoma cell lines.** Click chemistry was used on alkyne-myristate labeled cell lysates to determine overall protein myristoylation levels in: BL2 cells (**a**) and IM9 cells (**b**) treated for 1 h with 0.01–1.0 μM PCLX-001, myristoylation levels of a WT-Src-EGFP construct expressed in COS-7 cells (**c**) and, myristoylation of immunoprecipitated endogenous pp60-Src in IM9 cells following 1 h treatment with 1.0–10 μM of PCLX-001 (**d**). Fluorescence micrographs of COS-7 cells transfected with a WT-Src-EGFP (left), non-myristoylatable G2A-Src-EGFP mutant (center), and a WT-Src-EGFP construct treated with 10 μM PCLX-001 for 24 h (right) (**e**). Scale bars are equal to 10 μm. Endogenous Src protein levels in IM9 and BL2 cells treated with 1 μM PCLX-001 for 0–5 days measured by Western blotting (**f**). Western blotting of cleaved PARP-1 and cleaved caspase-3 in IM9, BL2, and Ramos cell lysates following 72 h incubation with 0–1.0 μM PCLX-001 (composite gels) (**g**). All data shown are representative of at least three independent experiments. GAPDH serves as loading control. Source data are provided as a Source Data file.

proteasome inhibition results in the stabilization of SFKs as suggested by their increased protein levels, it does not reverse the impact of PCLX-001 on ligand independent tyrosine phosphorylation, or overall SFK phosphorylation in BL2 cells (Fig. 3b) supporting the established notion that non-myristoylated SFKs are no longer functional because of their mislocalization and their inability to phosphorylate their substrates. Altogether, these results indicate that the myristoylation of SFKs is essential for both their activity and stability, and is required for downstream BCR signaling in lymphoma cells.

**PCLX-001 potently inhibits BCR survival signaling components.** Since PCLX-001 impacts SFK protein levels and ligand dependent BCR mediated tyrosine phosphorylation, we next evaluated its effects on other BCR mediated signaling

intermediates using two clinically approved BCR signaling inhibitors: dasatinib (a broad spectrum tyrosine kinase inhibitor) and ibrutinib (a BTK inhibitor) as controls[52]. Because BL2 cells were found to be most responsive to anti-human IgM BCR stimulation (Supplementary Fig. 8B), these cells were chosen as a model for studying PCLX-001-mediated effects on activated BCR signaling. BL2 cells treated with 0.1 or 1.0 μM PCLX-001 exhibit concentration dependent partial (at 24 h, Supplementary Fig. 8B) and near complete abrogation (at 48 h) of anti-IgM stimulated BCR mediated tyrosine phosphorylation (Fig. 4a, quantification in Supplementary Fig. 9). The overall reduction in tyrosine phosphorylation is more pronounced in BL2 cells treated with PCLX-001 than those treated with dasatinib or ibrutinib at the same concentrations. PCLX-001 treatment also reduces or abolishes levels of total Lyn, activated-phosphorylated-Lyn (Y396), as well as that of total BTK and activated-phosphorylated-BTK (Y223) in

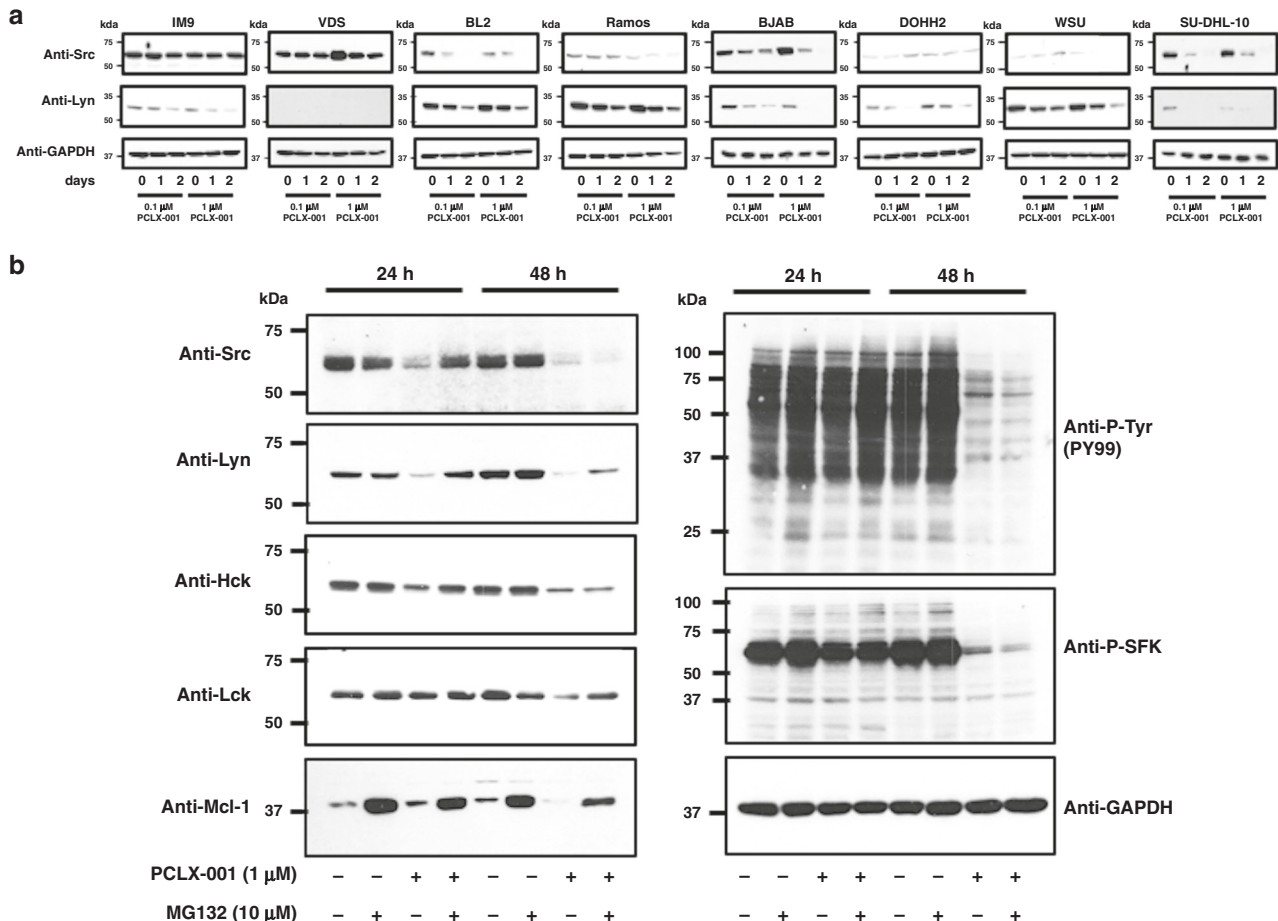

**Fig. 3 PCLX-001 treatment results in SFK instability and degradation by the proteasome in lymphoma cell lines.** Western blot for total Src and Lyn proteins in immortalized lymphocyte (IM9, VDS), BL (BL2, Ramos, BJAB), and DLBCL (DOHH2, WSU-DLCL2, SU-DHL-10) cell line lysates following 24–48 h of treatment with 0.1 μM or 1.0 μM PCLX-001 (**a**). After BCR ligation with anti-IgM, western blot for total Src, Lyn, Hck, Lck, Mcl-1, total phospho-tyrosine (PY99) and pan phosphorylated-SFK (P-SFK) protein levels in BL2 treated for 24–48 h with 1 μM PCLX-001 in the presence or absence of 10 μM of the proteasome inhibitor MG132 for the last 6 h (**b**). GAPDH serves as a loading control. All western blots shown are representative of three independent experiments. Source data are provided as a Source Data file.

BL2 cells (Fig. 4a). These findings were confirmed in several other lymphoma cell lines (Supplementary Fig. 10) and for several other SFKs including Src, Lck, Hck, and Fyn, as well as for activated-pan-phospho-SFKs in BL2 cells (Fig. 4a, Supplementary Fig. 11). Dasatinib and ibrutinib selectively inhibited their respective targets as measured using anti-P-Lyn, anti-P-SFKs, and anti-P-BTK antibodies (Fig. 4a).

PCLX-001 treatment also mediates the reduction of other myristoylated protein levels including the BCR signaling enhancer protein HGAL and Arf1 GTPases while dasatinib and ibrutinib have no effect on the levels of either of these proteins (Fig. 4b, c). Of note, the loss of HGAL protein was much faster than that of SFKs and Arf1 GTPase and the loss of HGAL protein levels is associated with a reduction in the phosphorylated and active form of SYK as expected[14,15] (Fig. 4b). Since the levels of both myristoylated HGAL and myristoylated small GTPase Arf1 are also diminished upon PCLX-001 treatment, the ability of PCLX-001 to promote the degradation of myristoylated proteins is therefore not restricted to myristoylated SFKs (Fig. 4).

BCR signaling ultimately converges on transcription factors involved in B-cell proliferation and survival including phospho-ERK (P-ERK), NFκB, c-Myc, and CREB[4,5]. Thus, we evaluated the effects of PCLX-001, dasatinib, and ibrutinib on these effectors at 0.1 and 1.0 μM for 48 h on BL2 cells. Consistent with

an impairment in BCR signaling, PCLX-001 reduces the levels of P-ERK, NFκB, c-Myc, and CREB in a concentration dependent manner with statistically significant decreases ($P < 0.05$) detected in phospho-ERK and NFκB levels (Fig. 4c, quantification in Supplementary Fig. 9). Again, these effects tend to be more marked in PCLX-001 treated cells than those treated with either dasatinib or ibrutinib. These findings, including decreased levels of Src, Lyn, pan-P-SFK, ERK, and P-ERK, are also observed in several other malignant lymphoma cell lines (Supplementary Fig. 10). We also show PCLX-001 treatment increased the levels of the ER stress pro-apoptotic marker Bip more than dasatinib and ibrutinib treatments leading to an overall increase apoptosis as measured by caspase-cleaved PARP1 (Fig. 4c). Therefore, the ability of PCLX-001 to promote the degradation of proteins is not restricted to its effects on myristoylated proteins such as SFKs, HGAL, and Arf1 but also includes effects on non-myristoylated proteins such as phospho-ERK and NFκB signaling downstream the BCR.

Early events in BCR signaling also culminate in the activation of phospholipase C$_\gamma$ and calcium mobilization in the cytosol. We demonstrate that PCLX-001 (1 μM) treatment of BL2 cells for 48 h potently inhibits anti-IgM BCR-induced calcium mobilization from intracellular stores using a fluorescent ratiometric Fura-2 Ca$^{++}$-chelator assay[53] (Supplementary Fig. 12). In addition to

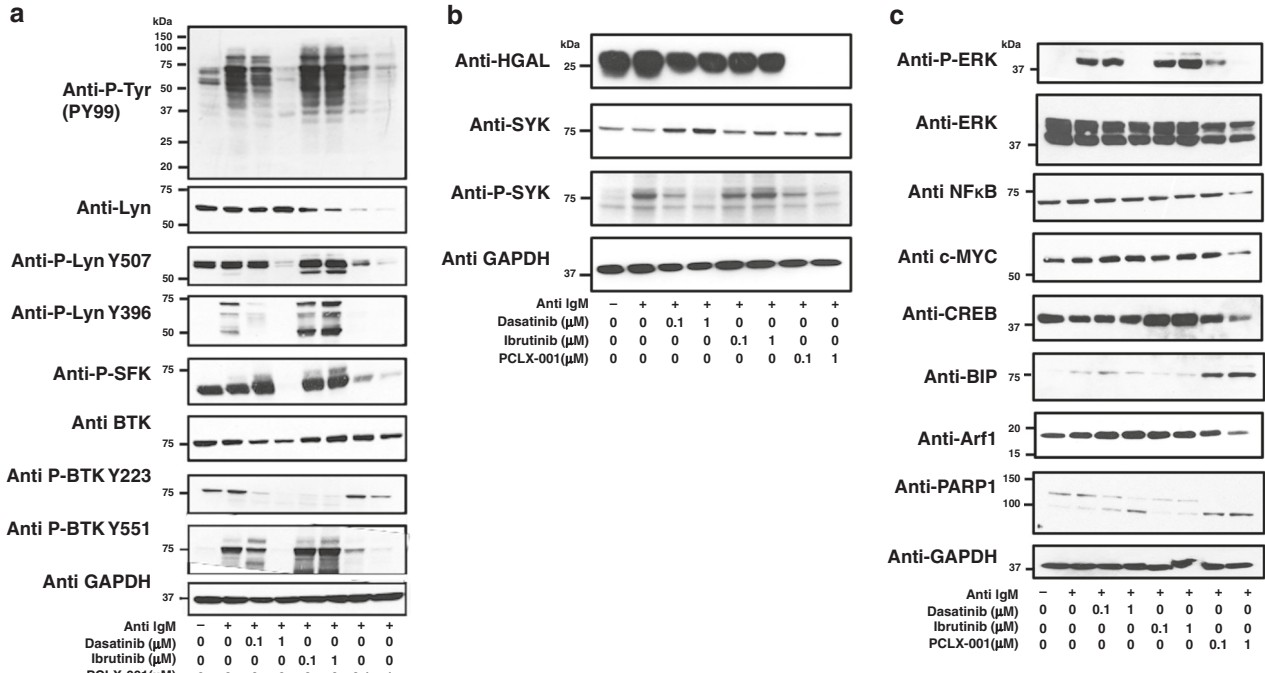

**Fig. 4 PCLX-001 treatment attenuates BCR downstream signaling events in BL2 lymphoma cells.** Western blot of BL2 cells treated for 48 h with 0.1 μM or 1.0 μM of dasatinib, ibrutinib or PCLX-001 to detect total tyrosine phosphorylation (P-Tyr), Lyn, Lyn phosphorylated on tyrosine 396 or 507, BTK, and BTK phosphorylated on tyrosines 223 or 551 (**a**), HGAL, SYK, phosphorylated SYK (P-SYK) (**b**) or ERK, phosphorylated ERK (P-ERK), NFκB, c-Myc, CREB, Arf-1, BIP, and PARP-1 (**c**). Western blots are representative of at least three independent experiments. GAPDH serves as a loading control. BL2 cells were activated with 25 μg/ml F(ab')₂ anti-human IgM for 2 min and processed for western blotting. All western blots shown are representative of three independent experiments. Source data are provided as a Source Data file.

drastically reducing the intensity of the calcium release peak, and similarly to dasatinib treatment, PCLX-001 delayed the calcium release process. Overall, PCLX-001 inhibited calcium mobilization more than either dasatinib and ibrutinib used at the same concentration. Of note, extended treatment of BL2 cells with PCLX-001 for 48 h interfered with calcium homeostasis and lead to increased basal levels of cytosolic calcium (Supplementary Fig. 12), perhaps contributing to ER calcium depletion and apoptosis. In all, our data indicate that PCLX-001 treatment effectively impairs BCR-mediated pro-survival signaling and induces apoptosis in lymphoma cells (Fig. 5).

Because PCLX-001, dasatinib and ibrutinib varied in potency and differentially affected downstream BCR signaling, we next compared the effects of these drugs on the overall viability of the lymphoma cell lines tested above. Dasatinib and ibrutinib treatments have minimal effect on BL2 (solid lines) and IM9 (dotted lines) cells following 48 and 96 h of treatment, whereas PCLX-001 kills malignant BL2 cells (solid line) at a substantially lower concentration than that required to kill benign, IM9 controls (dotted line) (Fig. 6a, b). Similar trends in cell viability are observed across all other cell lines with exception of SU-DHL-10, which was equally sensitive to both PCLX-001 and dasatinib (Fig. 6c, d). Of note, these treatments resulted in less than 25% cell death for PCLX-001 at 48 h and <5% for dasatinib and ibrutinib at either concentrations used (Fig. 6a and c). Importantly, the combination treatment of either dasatinib or ibrutinib at concentrations of 0.1 and 1.0 μM to PCLX-001 at 0.01, 0.1, and 1.0 μM does not further decrease viability suggesting that PCLX-001 effects are mediated upstream of dasatinib and ibrutinib targets (Supplementary Fig. 13). Altogether, PCLX-001 has the broadest spectrum of potency against malignant lymphoma cell lines at both 48 and 96 h in comparison

to dasatinib and ibrutinib, and is better at sparing benign, immortalized IM9 and VDS B-cell controls, demonstrating higher selectivity and an in vitro therapeutic window superior to that of two clinically approved drugs.

**NMT expression is altered in hematologic cancer cells.** While we still do not know why hematological cancer cells are more vulnerable to PCLX-001 than other cancer cell types, we think this might be related to alterations in *NMT1* or *NMT2* expression in hematological cancer cells. To substantiate this possibility, we performed in silico analyses of gene expression data from the Cancer Cell Line Encyclopedia[54]. We first find that the *NMT1* number of transcripts is about eight times ($2^3$) the number of *NMT2* transcripts in all cell lines on average, and second, that there is a heterogenous but significant reduction of *NMT2* expression in numerous hematological cancer cell lines in comparison to other types of cancer cell lines (Supplementary Fig. 15A, B). Expression of *NMT1* is relatively constant across the 1269 cell lines investigated with a slight but significant decrease in expression in breast and leukemia cancer cell lines while *NMT2* expression varies significantly amongst various cancers and also within a given cancer type (Supplementary Fig. 15C, D). The data also illustrate that while the expression of *NMT2* is higher in cancer cell lines of CNS, kidney and fibroblast origins there is a selective and significant reduction of *NMT2* expression in hematological cancers such as leukemia, lymphoma and myeloma (Supplementary Fig. 15D). Interestingly, the low *NMT2* expression levels seen in lymphomas, leukemia and other cell lines were not compensated by an increase in *NMT1* expression (Supplementary Fig. 15E). Altogether, we find a reduction in *NMT2* expression in hematologic cancer cell lines, which may account for their increased sensitivity to PCLX-001.

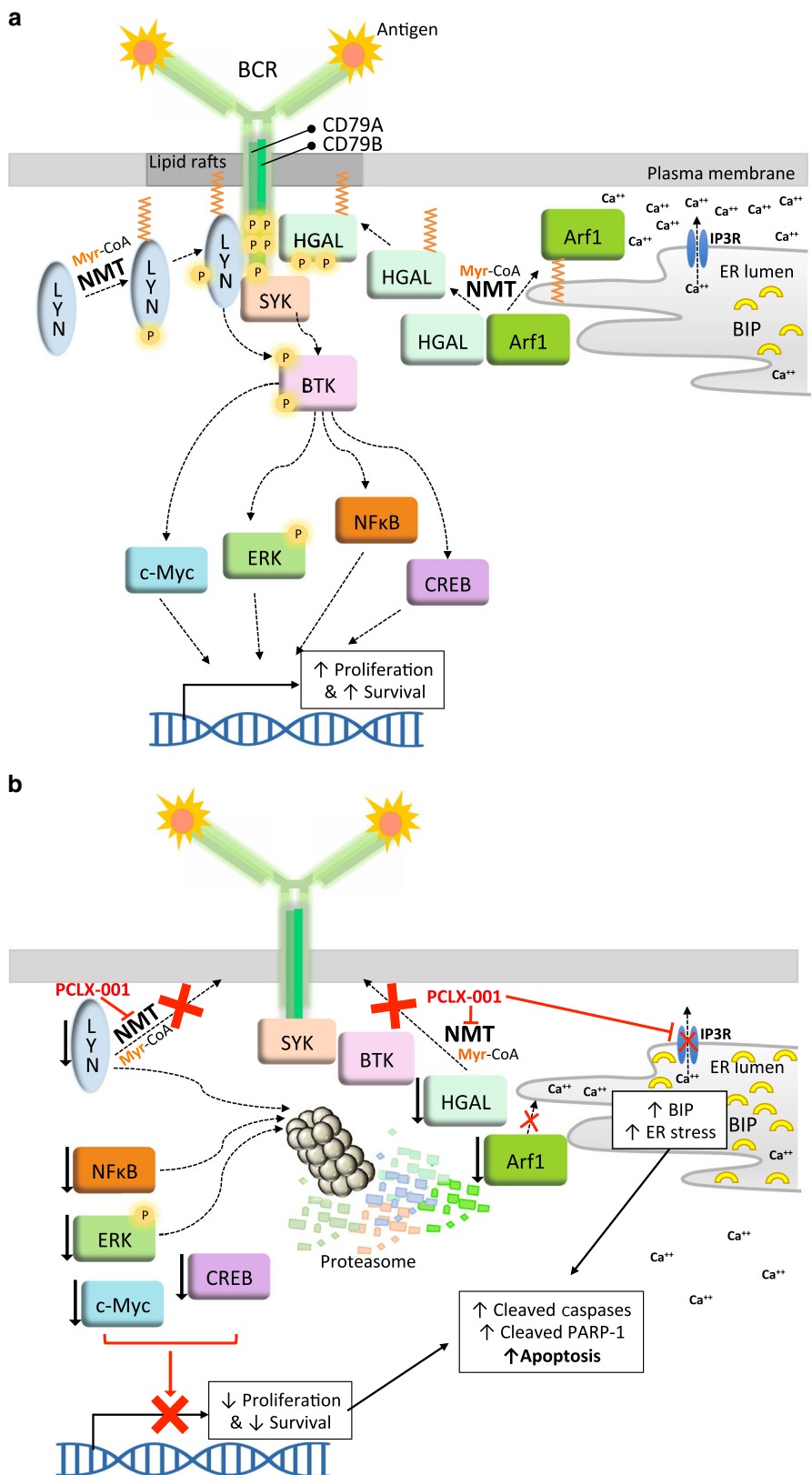

**PCLX-001 treatment has potent anti-tumor activity in vivo**. Based on lymphoma cell sensitivity to NMT inhibition in vitro, we investigated whether PCLX-001 could mitigate tumor progression in vivo in two murine lymphoma cell line-derived subcutaneous tumor xenograft models and used doxorubicin as a clinically approved drug reference. In mice bearing DOHH2 tumors, PCLX-001 demonstrates a significant tumoricidal effect when given daily at 20 mg/kg or every other day at 50 mg/kg ($P <$ 0.001) (Fig. 7a). At 50 mg/kg daily, PCLX-001 reduces tumor size by up to 70% by day 7 (average tumor size at day 7 = 44.0 ± 8.1 mm$^3$), but this was accompanied by significant weight loss, necessitating a 5-day treatment interruption (Supplementary

**Fig. 5 Model depicting proposed PCLX-001 mechanism of action in B cell lymphoma. a** Upon BCR activation, first the myristoylated SFK Lyn is recruited to the lipid raft domains of the plasma membrane containing the BCR, dephosphorylated Lyn at Y507 leads to its activation and autophosphorylation at Y396. This leads to the phosphorylation and activation of BTK at Y551 and Y223. Second, myristoylated HGAL is also recruited to the plasma membrane and phosphorylated thereby enhancing BCR signaling by stimulating SYK, BTK and the release of $Ca^{++}$ ions from the endoplasmic reticulum via the inositol-3-phosphate ion channel receptor (IP3R). Altogether these early signaling events lead to transcription activation by c-Myc, P-ERK, NFκB, and CREB. **b** The NMT inhibitor PCLX-001 prevents the myristoylation of Lyn-SFK (as well as other SFKs not shown in this model), HGAL and Arf1 thereby impeding the proper membrane targeting and function of these proteins. PCLX-001 treatment impedes calcium homeostasis by reducing the BCR mediated $Ca^{++}$ release from the ER and increasing basal $Ca^{++}$ levels in cells in addition to promote the degradation of both myristoylated (Lyn, HGAL, Arf1) and, surprisingly, non-myristoylated proteins (NFκB, P-ERK, c-Myc and CREB), some via the ubiquitin-proteasome pathway thereby further abrogating downstream BCR signaling and increasing ER stress leading to apoptosis and cell death.

Fig. 14A). Upon resuming treatment, a mean tumor growth inhibition (TGI) of 95% is observed by day 16. By comparison, doxorubicin treatment causes a 57% TGI and reduced body weight by up to 8% (Supplementary Fig. 14A). Importantly, treatment with PCLX-001 does not increase mortality at any dose (Supplementary Fig. 14B).

In mice bearing BL2 xenografts, PCLX-001 shows partial TGI at doses of 20 mg/kg daily reaching 42.5% tumor regression by day 9 ($P = 0.016$) (Fig. 7b). Furthermore, 50 or 60 mg/kg daily doses of PCLX-001 cause 100% tumor regression in nine of nine and seven of seven surviving mice, respectively, when administered for 13 days. Kaplan–Meier survival analysis of this xenograft model also shows that PCLX-001 doses between 20 and 50 mg/kg/day prolongs the survival of BL2 tumor bearing mice in comparison to untreated, vehicle controls (Supplementary Fig. 14D, E). Doxorubicin by contrast has no effect on BL2 tumor growth (Fig. 7b), and treatment was terminated at day 11 due to the adverse effects (Supplementary Fig. 14C). At the conclusion of treatment, we measured NMT activity[21] in BL2 tumor lysates and find it to be reduced in a PCLX-001 concentration-dependent manner ($P = 0.03$; Fig. 7c) showing that PCLX-001 acts on target in vivo.

Because cell line derived xenografts lack the complexity of human tumors, we dissected and propagated a DLBCL lymphoma derived from patient DLBCL3 whose cancer was refractory to multiple lines of chemotherapy including CHOP, RICE, intrathecal methotrexate/cytarabine, and DHAP (Supplementary Table 2) to establish a patient-derived xenograft model in NODscid mice. Treatments were assessed in groups of 8 mice each. A 20 mg/kg subcutaneous daily dose of PCLX-001 treatment for 21 days results in 66% TGI ($P < 0.001$; Fig. 7d). This dose was then increased to 50 mg/kg daily in another set of mice for two 9-day periods separated by a 3-day treatment interruption to allow the mice to recover from ~15% loss of body weight (Supplementary Fig. 14F). Following this higher dose regimen, PCLX-001 administration results in complete tumor regression in 6 of 7 surviving mice at day 13 (Fig. 7d) with one mouse with no detectable tumors dying at day 11 (Supplementary Fig. 14G). Surgically removed tumors from vehicle-control and PCLX-001 treated mice confirm a concentration-dependent reduction in overall tumor size following 21 days of PCLX-001 treatment (Fig. 7e) concomitant with increased in apoptosis (increased cleaved caspase-3; Fig. 7f) and reduction in cell proliferation (as determined by Ki-67 analysis; Fig. 7g). Thus, PCLX-001 treatment induces apoptosis and cell-cycle arrest in a patient-derived lymphoma tumor in vivo in a dose-specific manner. The effect of doxorubicin treatment could not be assessed due to severe drug toxicity and death in the majority of tumor bearing mice within the first 7 days of the experiment.

**Mice tolerate PCLX-001 at efficacious dose levels.** Mice tolerated PCLX-001 at efficacious doses without specific end-organ toxicity. All mice treated with PCLX-001 survived the first

xenograft study (Fig. 7a), while some mice treated with PCLX-001 at higher dose levels died in the other two studies (Fig. 7b, d). Neither the clinical pathology nor anatomic pathology evaluations identified the cause of death. Findings suggesting toxicity were seen in two studies. Of three mice bearing BL2 xenografts and given PCLX-001 at 50 mg/kg daily with a short treatment holiday, all had lower-than-normal neutrophil and lymphocyte counts at the end of the dosing period, and one also had lower-than-normal monocyte and platelet counts. In mice bearing DLBCL lymphoma cell xenografts and given PCLX-001 at 20, 50, or 60 mg/kg daily, signs of ill health (e.g., rough and scruffy coats; piloerection) were noted in most mice at all dose levels, and dehydration and weight loss were noted at 50 and 60 mg/kg daily (Supplementary Tables 3–8).

Dose ranging toxicology studies in rat and dog have been performed and reported[55], and formal GLP toxicology studies in these species are nearing completion in preparation for regulatory review for human clinical trials.

Altogether, our results demonstrate that PCLX-001 treatment inhibits the growth of lymphomas in vivo, including the complete regression of disease refractory to other clinically approved treatments and thus establishes a proof-of-concept for the use of a bona fide NMT inhibitor such as PCLX-001 in cancer.

**Discussion**
Herein, we report the discovery that hematological cancer cells, particularly B-cell lymphomas, are highly sensitive to myristoylation inhibition by the novel pan-NMT inhibitor PCLX-001. While the concept of killing cancer cells with a NMT inhibitor has been proposed and tested on small scales[39,43,56–59], to our knowledge this work represents the original investigation of the breadth of efficacy of this approach across hundreds of cancer cell lines. We demonstrate that cancer cells can be selectively killed by a NMT inhibitor at concentrations lower than that required to kill and inhibit the proliferation of immortalized and normal cells (Fig. 1e–h, Supplementary Figs. 3 and 4). In the absence of additional cytotoxicity assays in more normal cell types, based on the benefit/risk for this therapeutic indication, it is acceptable and not unusual that some normal tissues (e.g., blood cells including PBMCs) are effected at efficacious doses. Our data indicate a large enough therapeutic window critical to support the development of PCLX-001 as a potential cancer treatment. In addition to inhibiting the myristoylation of a large number of myristoylated proteins in B lymphoma cells (Fig. 2a, b), we demonstrate that PCLX-001 is especially efficient at inhibiting BCR signaling, which is the main lymphoma pro-survival pathway in these cells[4–8]. In addition, the PCLX-001 BCR signaling inhibition is superior to that of clinically approved SFK inhibitor dasatinib and the BTK inhibitor ibrutinib. This may explain in part why PCLX-001 also has the broadest spectrum of potency against malignant lymphoma cell lines in vitro. We also show PCLX-001 inhibits the myristoylation of SFKs, HGAL, and Arf1 and increases their degradation rates, but also unexpectedly promotes the

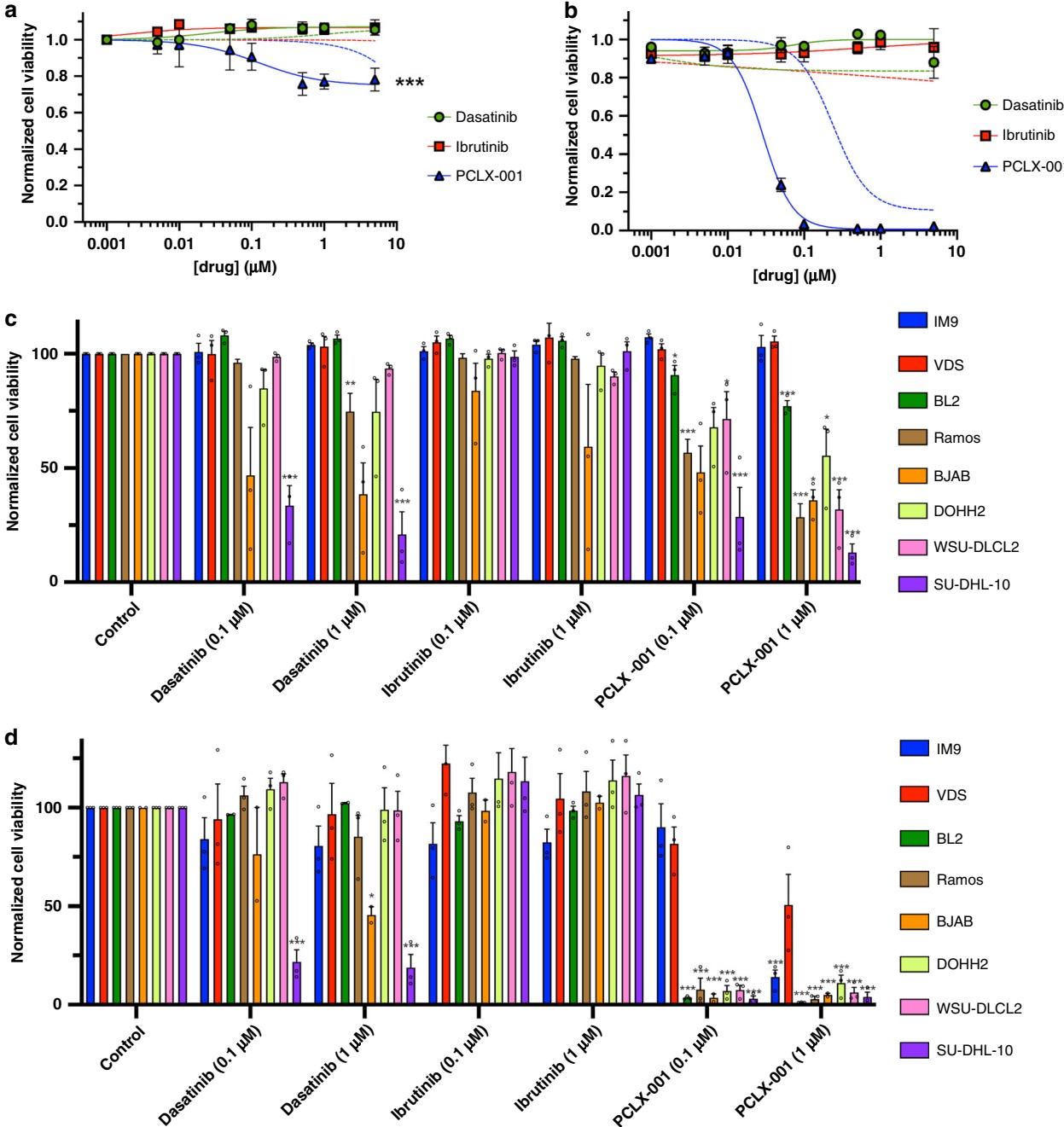

**Fig. 6 PCLX-001 selectively kills hematological cancer cells relative to benign lymphocytes in comparison to dasatinib and ibrutinib.** Cell viability curves of BL2 (solid lines) and IM9 cells (dotted lines) treated for 48 h (**a**) or 96 h (**b**) with 0.001–5 μM dasatinib, ibrutinib, or PCLX-001(2way Anova, Tukey's multiple comparisons test, $P < 0.0001$) . Normalized cell viability of immortalized lymphocyte (IM9, VDS), BL (BL2, Ramos, BJAB), and DLBCL (DOHH2, WSU-DLCL2, SU-DHL-10) cell lines treated with 0.1 μM or 1.0 μM of dasatinib, ibrutinib or PCLX-001 for 48 h (**c**) and 96 h (**d**). Cell viability for all experiments was measured using Calcein assay and is an average of three independent experiments. (Ordinary one-way Anova, Dunnett's multiple comparisons test) Errors bars depict s.e.m. Source data are provided as a Source Data file.

degradation of non-myristoylated pro-survival BCR mediators including P-ERK, NFκB, c-Myc, CREB, and perhaps even BTK (Fig. 4a). PCLX-001 treated cells still remained at least 75% viable at concentrations that are becoming cytotoxic. Whether the lower downstream signaling protein levels correspond to a reduction in gene transcription or increased protein degradation in dying cells is not known. Furthermore, PCLX-001 also reduces BCR-mediated calcium mobilization causing apoptosis selectively in B cell lymphoma cells (Fig. 5). The mechanism linking the loss of

myristoylation to alterations in calcium homeostasis and inhibition of BCR mediated calcium release is not known.

Increased ER stress is a pro-apoptotic phenomenon previously shown in cells treated with another NMT inhibitor[59]. We postulate the inhibition of myristoylation of the Arf1 GTPase, whether at its N-terminal glycine residue or nearby lysine residue[36,37], interferes with its membrane targeting and impairs vesicle trafficking thereby detrimentally affecting chronic/tonic or antigen dependent BCR signaling. Loss of proper Arf1 functionality at the

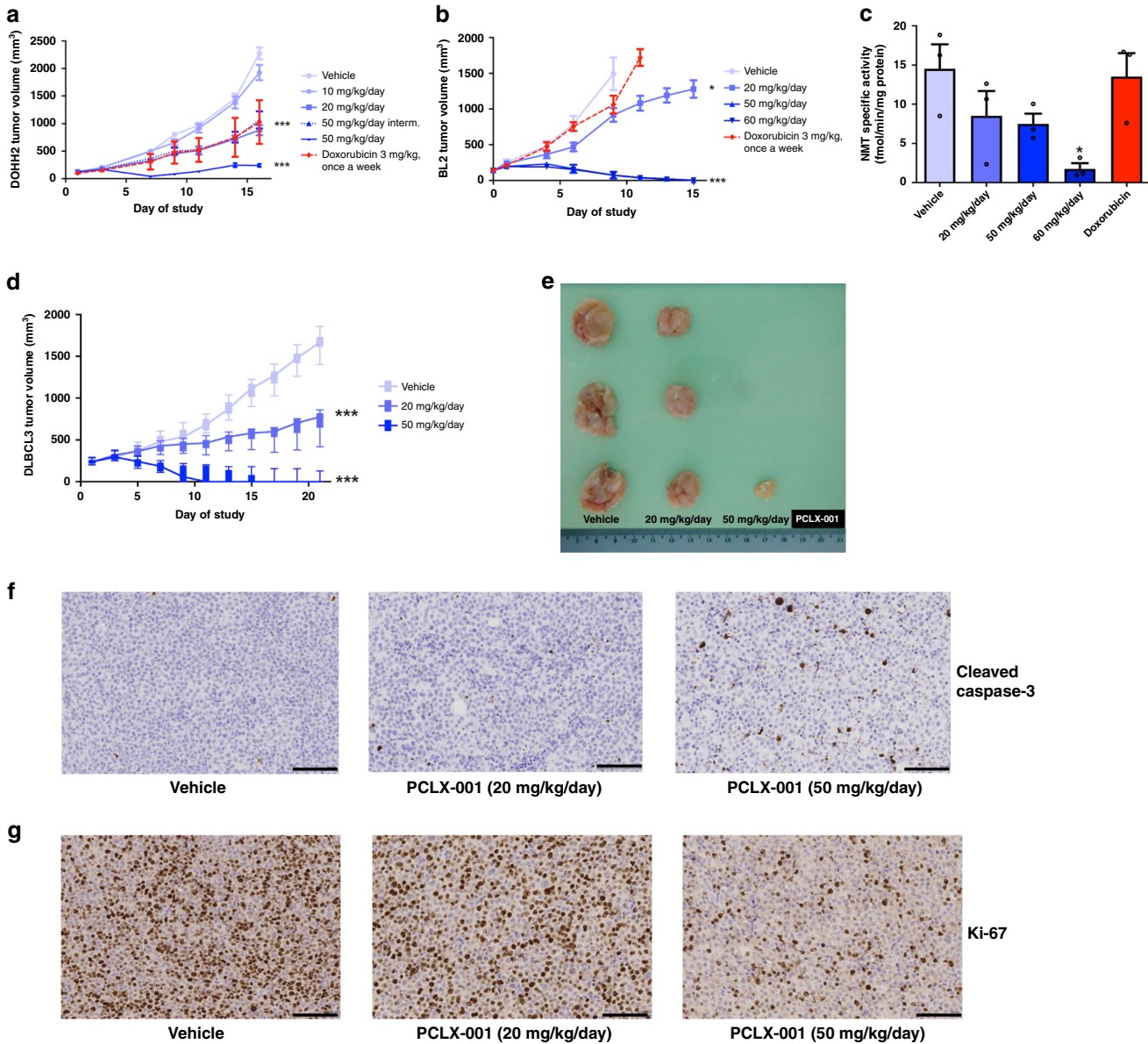

**Fig. 7 PCLX-001 treatment reduces tumor volumes and leads to complete tumor regression in B-cell lymphoma xenograft models.** Dose–response curves for murine subcutaneous xenografts derived from cell lines measuring the size of DOHH2 (**a**) and BL2 (**b**) tumors as a function of time. Error bars represent the standard deviation of average tumor volumes ($n = 10$ per group). Average total NMT specific activity assessed as previously described[21] in BL2 tumor samples from mice treated with PCLX-001, doxorubicin, or vehicle alone at the indicated doses. Tumor extracted from mice treated with 60 mg/kg/day have reduced NMT specific activity as compared to vehicle (paired $t$-test, $P = 0.0425$). Error bars represent s.e.m. (**c**). Dose–response curve for the murine xenograft derived from patient DLBCL3. Datapoints represent average tumor volumes in all surviving animals. Error bars represent the standard deviation in the average tumor volumes (**d**). ***Indicates a significant difference in response rate between animals which received 20 mg/kg/day and 50 mg/kg/day of PCLX-001 ($P < 0.0001$, $n = 8$ per group). Representative tumors from mice with patient-derived DLBCL3 xenografts (**e**). Representative IHC staining for cleaved caspase-3 (**f**) and Ki-67 (**g**) in the above DLBCL3 patient xenograft tumor samples. Scale bars equal to 100 μm. Source data are provided as a Source Data file.

ER may also explain in part the increase in ER stress marker Bip[60] upon PCLX-001 treatment (Fig. 4c).

The loss of lipid raft localized myristoylated Lyn (and other SFKs) and HGAL proteins in PCLX-001 treated cells further highlights the importance of these membrane domains in proper BCR signaling[9,10,12–15] (Fig. 5). Furthermore, PCLX-001-mediated myristoylation inhibition of SFKs not only abrogates their membrane targeting but also promotes their degradation via the ubiquitin-proteasome system as MG132 treatments resulted in near complete recovery of SFK levels (Fig. 3b). While ubiquitination and degradation of protein tyrosine kinases by the

Casitas B lineage lymphoma-family of E3 ubiquitin ligases[61,62] is a normal part of the signal attenuation in B cells, an N-terminal glycine residue has also recently been shown to be a destabilizing factor for proteins, representing a highly selective novel class of N-degron[63]. Indeed, in their report, Timms et al. demonstrate that unmyristoylated proteins including Lyn, Fyn and Yes, exposing their N-terminal glycine residue are selectively degraded by the N-terminal glycine specific Cullin RING Ligase 2 (CRL2)-ZYG11B/ZER1 N-degrons–ubiquitin–proteasome system[63]. This system is highly selective for proteins with a N-terminal glycine residue since substitutions of glycine for any other amino acid led

to a substantial stabilization of the resulting proteins[63]. This newly described N-degron system[63,64] may therefore contribute to the faster degradation of unmyristoylated proteins seen in malignant lymphoma cell lines treated with PCLX-001 such as SFKs, HGAL, and Arf1 (Fig. 4). It might also explain in part why non-myristoylatable Gly2Ala-Src tyrosine kinase mutant and Gly2Ala-HGAL were previously shown to be more stable than their myristoylated counterpart proteins[65,66] since the artificial N-terminal alanine (Ala) residue would prevent the promotion of degradation by the glycine (Gly) residue specific CRL2-ZYG11B or CRL2-ZER1 N-degrons. Thus, we propose a model for the mode of action of PCLX-001 in B-cell lymphoma whereby inhibition of myristoylation of SFKs (or other proteins including HGAL and Arf1) results not only in a loss of membrane targeting but also in a loss of their protein levels and thus function, via the ubiquitin–proteasome system (Fig.3B), thereby dampening the propagation of BCR signals (Fig. 5). Interestingly, NMT1 was found to be phosphorylated by Lyn, Fyn, and Lck SFKs and that phosphorylation of NMT1 was necessary for myristoylation activity since a non-phosphorylatable Y100F-NMT1 mutant lost 98% of its catalytic activity[67]. Therefore, the PCLX-001 mediated loss of SFKs could further reduce NMT1 activity in B lymphoma cells thereby potentiating the loss of pro-survival signals and apoptosis.

In addition to the effects depending on myristoylated SFKs, HGAL, and Arf1 proteins, given that there are hundreds of known myristoylated proteins, PCLX-001-mediated effects on lymphoma cell viability likely also occur via the loss of functionality of other myristoylated proteins. Although we still do not know why hematological cancer cells are more vulnerable to PCLX-001 than other cancer cell types, we think this is possibly related to altered expression of either NMT1 or NMT2. Analysis of CCLE NMT1 or NMT2 expression data (Supplementary Fig. 15) reveals that in addition to be overexpressed in some cancers (aka the current dogma), NMT expression levels are actually lower in other cancers, many of which are of hematological origin. Altogether, these observations suggest a possible link between the reduction in the number of NMT enzyme targets in hematological cancer cells and the sensitivity of these cells to PCLX-001. Whether altered NMT levels impact on the sensitivity of hematological cancer cells on their own or possibly work in combination with variations in the individual myristoylated proteomes of hematological cancer cells, and, the cell-specific reliance of these cells on various myristoylated proteins for survival is not known. While these possibilities are currently under further investigation in our laboratory, the potential importance of NMT activity to lymphoma cell survival was confirmed in a genome-wide Cas9-Crispr screen in which NMT1 ranked amongst the most critical survival factors in lymphoma cell lines[68]. In addition, our cancer cell line screen results suggest potential for a broader application of PCLX-001 to treatment of leukemia and myeloma, as well as certain solid tumors such as breast and lung cancers.

While PCLX-001 is only marginally efficacious at the tolerated dose of 20 mpk [~66% tumor reduction (Fig.7)], we show that it effectively inhibits tumor cell growth in vivo resulting in either major or complete regression of disease in three human lymphoma xenograft models at the 50 mpk efficacious dose, including complete response in a lymphoma refractory to CHOP, Rituximab, and other salvage therapies.

In conclusion, we establish our initial proof-of-concept that a small molecule NMT inhibitor, PCLX-001, potently and selectively inhibits the growth of a wide spectrum of cultured cancer cells in vitro, with particularly pronounced effects in cells derived from hematologic cancers including B-cell lymphoma due to the loss of BCR-mediated signaling events, their main source of pro-

survival signals[4–8]. Together with the striking efficacy of PCLX-001 in pre-clinical models of B-cell lymphoma in vivo, these findings support the ongoing development and potential clinical trials of PCLX-001 and related NMT inhibitors as therapies for B cell lymphoma and possibly other cancers.

## Methods

**Antibodies and materials**. Rabbit anti-PARP-1 (1:5000, affinity purified polyclonal#EU2005, lot 1), anti-GAPDH (1:5000, affinity purified polyclonal, #EU1000, lot 1), and anti-GFP (1:10,000, affinity purified, #EU1, lot B3-1) were from laboratory stock and are available through Eusera (www.eusera.com). Our affinity purified rabbit anti-GFP is also available as Ab6556 from Abcam (Cambridge, MA). Rabbit monoclonal anti-Src (1:2000, clone 32G6, #2123, lot 5), Lyn (1:2000, clone C13F9, #2796, lot 4), P-Lyn Y507 (1:5000, polyclonal, #2731, lot 5), Fyn (1:2000, polyclonal, #4023, lot 3), Lck (1:2000, clone D88, #2984, lot 4), Hck (1:2000, clone E1I7F, #14643, lot 1), c-Myc (1:10,000, clone D3N8F, #13987, lot 5), ERK (1:2000, clone 4695, #9102, lot 27), P-ERK (1:5000, clone 3510, #9101, lot 30), P-SFK (1:10,000, clone D49G4, #6943, lot 4), BTK (1:2000, clone D3H5, #8547, lot 13), P-BTK Y223 (1:5000, clone D9T6H, #87141, lot 1) SYK (1:2000, clone D3Z1E, #13198, lot 5), P-SYK Y525/526 (1:5000, clone C87C1, lot 18), and anti-cleaved caspase-3 (1:1000, clone 5A1E, #9664, lot 20) were purchased from Cell Signaling Technologies. Rabbit monoclonal anti-BIP (1:2000, polyclonal, ADI-SPA-826) was purchased from Enzo Life Sciences. Rabbit anti-Mcl-1 (1:2000, clone Y37, #32087, lot GR119342-5), NFκB (1:2000, clone E379, #32536, lot GR3199609-2), P-Lyn Y396 (1:5000, polyclonal, #226778, lot GR3195652-5) were purchased from Abcam (Cambridge, MA). Mouse monoclonal anti-p-Tyr (1:10,000, PY99, sc-7020, lot I2118) antibody was purchased from Santa Cruz Biotechnology. Mouse anti human HGAL was purchased at eBioscience (1:10,000, clone 1H1-A7, #14-9758-82, lot E24839-101). Rabbit polyclonal anti-ARF-1 antibody (1:2000, polyclonal, #PA1-127, lot TK 279638) was purchased from ThermoFisher Scientific. Enhanced chemiluminescence (ECL) Prime Western blotting detection kits were purchased from GE Healthcare. Clarity ECL western blotting substrate was from Bio-Rad. Goat anti-human IgM (μ chain) (70-8028-M002, lot S728028002001) was purchased from Tonbo biosciences. Goat F(ab')2 anti-human IgM was purchased from BioRad (STAR146, lot 152684). Rabbit Anti-human Src antibody from Sigma-Aldrich (polyclonal, Ab-529, lot 871521168) was used for immunoprecipitation. Doxorubicin hydrochloride was from Pfizer. Dasatinib and ibrutinib were from ApexBio Technology. PCLX-001 was identified as DDD86481 by Drs. David Gray and Paul Wyatt (University of Dundee, Scotland, UK)[38,69]. All chemicals were of the highest purity available and purchased from Sigma-Aldrich, unless indicated otherwise.

**Cell culture**. IM9, Ramos, SU-DHL-10, and COS-7 were purchased from ATCC. BL2, DOHH2, WSU-DLCL2, and BJAB were purchased from DSMZ (Germany). Ramos and BL2 were kind gifts of Drs. Jim Stone and Robert Ingham of University of Alberta. VDS isolation was described in Tosato et al.[47]. VDS, BJAB, and SU-DHL-10 were kind gifts of Dr. Michael Gold of the University of British Columbia. HUVEC cells (pooled from up to four umbilical cords) were purchased from PromoCell. All cell lines identity was confirmed by STR profiling at The Genetic Analysis Facility, The Centre for Applied Genomics, The Hospital for Sick Children, Peter Gilgan Centre for Research and Learning, 686 Bay St., Toronto, ON, Canada M5G 0A4 (www.tcag.ca). Cell lines were tested regularly for mycoplasma contamination using MycoAlert Plus Mycoplasma Detection Kit (Lonza, ME, USA). All cell lines tested negative for mycoplasma contamination. All cell lines were maintained in RPMI or DMEM medium supplemented with 5–10% fetal bovine serum, 100 U/ml penicillin, 0.1 mg/ml streptomycin, 1 mM sodium pyruvate, and 2 mM L-glutamine. HUVEC cells (pooled from up to four umbilical cords) were purchased from PromoCell and cultured in Endothelial cell growth media with Insulin-like Growth Factor (Long R3 IGF) and Vascular Endothelial Growth Factor and maintained at passages lower than seven. All cell lines were maintained at 37 °C and 5% CO$_2$ in a humidified incubator and routinely checked for the presence of contaminating mycoplasma. Please see Supplementary Table 3 for cell line names, types and histology. For transfections, adherent cells COS-7 cells were transfected using X-tremeGENE9 DNA (Roche) transfection reagent according to manufacturer's instructions. For BCR activation experiments, cells were incubated with 25 μg/ml of Goat F(ab')$_2$ anti-human IgM (or anti-human IgM (μ chain) showing identical BCR activation properties) for 2 min and the activation was stopped by the addition of 1 mM vanadate (Bio Basic Inc) solution in PBS.

**Lysis of cells**. Cells were harvested, washed in cold PBS, and lysed in 0.1% SDS-RIPA buffer (50 mM Tris-HCl pH 8.0, 150 mM NaCl, 1% Igepal CA-630, 0.5% sodium deoxycholate, 2 mM MgCl$_2$, 2 mM EDTA with 1× complete protease inhibitor; (Roche Diagnostics) by rocking for 15 min at 4 °C. The lysates were centrifuged at 16,000 g for 10 min at 4 °C, and the post-nuclear supernatant was collected.

**Immunoblotting, immunoprecipitation, and metabolic labeling of cells with alkyne-myristate**. Protein concentrations were determined by BCA assay (Thermo

Scientific) according to manufacturer's instructions. Samples were prepared for electrophoresis by the addition of 5× loading buffer and boiled for 5 min. If not stated otherwise, 30 μg of total protein per lane is loaded on a 12.5% acrylamide gels. After electrophoresis, gels are transferred onto 0.2 μM nitrocellulose membrane (Bio-Rad) thereafter probed with antibodies as described in materials section. Peroxidase activity is revealed following the procedure provided for the ECL Prime Western Blotting Detection Reagent (GE Healthcare, PA, USA).

Immunoprecipitation was performed as previously described in Yap et al.[47]. Briefly, cells are washed with cold PBS, harvested, and lysed with cold EDTA-free RIPA buffer (0.1% SDS, 50 mM HEPES, pH 7.4, 150 mM NaCl, 1% Igepal CA-630, 0.5% sodium deoxycholate, 2 mM MgCl₂, EDTA-free complete protease inhibitor (Roche)) by rocking for 15 min at 4 °C. Cell lysates are centrifuged at 16,000 g for 10 min at 4 °C and the post-nuclear supernatants are collected. EGFP fusion proteins or endogenous c-Src non-receptor tyrosine kinase (Src) were immunoprecipitated from approximately 1 mg of protein lysates with affinity purified goat anti-GFP (www.eusera.com) or rabbit anti-Src antibody (Sigma, Ab-529, lot 871521168) by rocking overnight at 4 °C. Pure proteome protein G magnetic beads (Millipore) were incubated with immunoprecipitated proteins for 2 h and extensively washed with 0.1% SDS-RIPA, re-suspended in 1% SDS in 50 mM HEPES, pH 7.4 and heated for 15 min at 80 °C. The supernatants containing the immunoprecipitated proteins were collected for Western blot analysis or click chemistry.

IM9, BL2, and COS-7 cells were treated with PCLX-001 for 1 h and cells were then labeled with 25 μM ω-alkynyl myristic acid 30 min before harvesting at each time point. Protein from the resulting cell lysates were reacted with 100 μM azido-biotin using click chemistry and processed as described in Yap et al.[47] and Perinpanayagam et al.[33].

**Viability of cells treated with PCLX-001, dasatinib, and ibrutinib**. IM9, VDS, BL2, Ramos, BJAB, DOHH2, WSU-DLCL2, and SU-DHL-10 cells (1 × 10⁵ cells) were grown in six-well plates in 4 ml media/well and incubated with increasing concentrations of PCLX-001, dasatinib, and ibrutinib for up to 96 h. Viability of cells treated with PCLX-001 was measured by CellTiter-Blue Cell Viability Assay (Promega) or with calcein AM staining (Life Technologies) according to the manufacturer's instructions on a Cytation 5 plate reader (Biotek, Winooski, VT). Calcein assay consists of measuring the cell viability ratio (live cells/total cells and expressed as % viability). Cells were stained with the Nuclear-ID Blue/Red cell viability reagent (GFP-certified, Enzo Life Sciences) to identify total cells, and dead cells while live cells were stained with Calcein AM (Life Technologies) according to manufacturer's instructions. Cell count was performed using a Cytation 5 Cell Imaging Multi-Mode Reader (Biotek Instruments, Inc.) and analyzed by Biotek Gen5 Data Analysis software (version 2.09).

Cell viability was also measured using the Horizon (St. Louis, MO) platform. Cells were seeded in growth media in black 384-well tissue culture treated plates at 500 cells per well. Cells are equilibrated in assay plates via centrifugation and placed in incubators at 37 °C for 24h before treatment. At the time of treatment, a set of assay plates (which do not receive treatment) are collected and ATP levels are measured by adding ATPLite© (PerkinElmer, Waltham, MA). These Tzero ($T_0$) plates are read using ultra-sensitive luminescence on Envision plate readers. Assay plates are incubated with compound for 96 h (except where noted in Analyzer) and are then analyzed using ATPLite©. All data points are collected via automated processes and are subject to quality control and analyzed using Horizon's Chalice Analyzer proprietary software (1.5). Assay plates were accepted if they passed the following quality control standards: relative raw values were consistent throughout the entire experiment, Z-factor scores were greater than 0.6 and untreated/vehicle controls behaved consistently on the plate. Horizon utilizes Growth Inhibition (GI) as a measure of cell growth. The GI percentages are calculated by applying the following test and equation:

$$\text{If } T < V_0 : 100 * \left(1 - \frac{T - V_0}{V_0}\right),$$

$$\text{If } T \geq V_0 : 100 * \left(1 - \frac{T - V_0}{V - V_0}\right),$$

where $T$ is the signal measure for a test article, $V$ is the untreated/vehicle-treated control measure, and $V_0$ is the untreated/vehicle control measure at time zero (also colloquially referred as $T_0$ plates). This formula is derived from the Growth Inhibition calculation used in the National Cancer Institute's NCI-60 high throughput screen. 100% GI therefore represents complete growth inhibition (cytostasis) while 200% GI represents complete cell death.

Cell viability was also measured using the Oncolines (Netherlands Translational Research Center B.V.) platform. Cells were diluted in the corresponding ATCC recommended medium and dispensed in a 384-well plate, depending on the cell line used, at a density of 200–6400 cells per well in 45 μl medium. For each used cell line the optimal cell density is used. The margins of the plate were filled with phosphate-buffered saline. Plated cells were incubated in a humidified atmosphere of 5% CO₂ at 37 °C. After 24 h, 5 μL of compound dilution was added and plates were further incubated. At t = end, 24 μL of ATPlite 1Step™ (PerkinElmer) solution was added to each well, and subsequently shaken for 2 min. After 10 min of incubation in the dark, the luminescence was recorded on an Envision multimode reader (PerkinElmer).

Finally, 3rd breadth of PCLX-001 efficiency screen (Supplementary Fig. 2) was performed using the ChemPartner platform (Shanghai, China). One hundred and thirty one cell lines were seeded in 96-well plate, black wall, tissue culture treated (from Corning, Cat.3904) and cultured following ATCC formulation. Cell viability after 72 and 144 h was measured using Cell Titer Blue Viability Assay (from Promega, Cat. G8081, Lot. No. 0000190181) and fluorescence at 560/590 nm was recorded with Enspire (PerkinElmer). EC₅₀ was calculated using XLfit software (5.5).

**Cell proliferation assay**. Proliferation of cells was measured by imaging and counting after digital phase contrast picture transformation for better accuracy. 2 × 10⁵ cells were cultured in six-well plates in 4 ml of culture media and incubated with increasing concentration of PCLX-001. After homogenization, 50 μl of culture was transferred into a high binding clear glass bottom ½ area 96 well plate (Greiner bio-one). Total well area was imaged in bright field (12 stitched pictures) using a Cytation 5 Cell Imaging Multi-Mode Reader (Biotek Instruments, Inc.) and transformed into a single digital phase contrast picture. Total cell counts were performed daily for up to 4 days (Biotek Gen5 Data Analysis software 2.09).

**Intracellular calcium measurements**. Cytosolic free calcium concentration measurements were performed in BL2 lymphoma cells incubated for 24 or 48 h with 1 μM PCLX-001, dasatinib or ibrutinib using PTI fluorometer (Photon Technology International) using adapted previously described protocol[53]. 10 × 10⁶ cells are suspended in fresh media with 8 μM Fura-2 AM (Molecular Probes) and 1 mM CaCl₂ for 30 min, washed and resuspended in media supplemented with calcium for an additional 15 min. Cells are then washed and resuspended in warm Krebs Ringer solution (10 mM HEPES pH 7.0, 140 mM NaCl, 4 mM KCl, 1 mM MgCl₂ and 10 mM glucose) and placed in a four-sided clear cuvette. Prior to activation, the free cytoplasmic calcium was chelated with 0.5 mM EGTA for 1 min. BCR receptor dependent calcium release is activated by the addition of 10 μg/ml Goat F(ab')₂ anti Human IgM (BioRad). Following, Thapsigargin (300 nM) was used to show BCR-independant and irreversible Ca²⁺ release from the endoplasmic reticulum. Ca2+ concentrations were calculated with the following equation:

$[Ca^{++}] = Kd (R–Rmin)/(Rmax–R)$ with R = Fluorescence Intensity at 340 nm divided by fluorescence intensity at 380 nm, Rmax = fluorescence measured following Ionomycin (7.5 μM) and CaCl₂ (12 mM) addition, Rmin = fluorescence measured following EGTA (32mM), Tris (24 mM) and Triton X-100 (0.4%) and Kd = 224 (at 37 °C for Fura-2 AM).

Results shown are representative of multiple replicates of the experiment ($n = 6$ for PCLX-001 incubation, $n = 3$ for Dasatinib and Ibrutinib).

**Isolation of PBMC and lymphocytes and cell viability assay**. Two healthy human research volunteers were recruited for PBMC and lymphocytes isolation from a 20 ml blood collection (patient #1: male, 34 years old, no diagnosis, no treatment; patient #2: male, 54 years old, no diagnosis, no treatment). Study protocol was approved by the Health Research Ethics Board of Alberta Cancer Committee (Study title: Evaluations of Fatty AcylTransferases (FATs) in fresh blood and blood forming cells; HREBA.CC-17-0624).

Mononuclear cells were isolated from peripheral blood by density gradient centrifugation using Ficoll-Paque (GE Healthcare, PA, USA). Lymphocytes were isolated from whole blood samples using EasySep™ lymphocyte isolation kit (Stemcell Technologies, Vancouver, BC, Canada) as per manufacturer's instructions. PBMC and lymphocytes were cultured in RPMI medium with 10% FBS, 100 U/ml penicillin, 0.1 mg/ml streptomycin. Cells were plated at a concentration of 2 × 10⁶ cells/ml. After incubation with 0.001–10 μM PCLX-001 for 96 h, cell viability was measured by using CellTiter-Fluor™ viability assay (Promega, Madison, WI, USA).

**Immunohistochemistry**. COS-7 cells were cultured plated on Poly-d-Lysine-coated 35-mm glass-bottom dishes (MatTek Corporation, Ashland, MA, USA) and transiently transfected with the indicated fluorescently tagged proteins using X-tremeGENE9 DNA (Roche) as recommended by the suppliers. Images were acquired using a Zeiss Observer Z1 microscope and Axiovision software (Axiovision, version 4.8). B-cell lymphomas were fixed in formalin, embedded in paraffin, cut into 5 mm sections with a microtome, mounted on Superfrost Plus slides (Fisher Scientific), deparaffinized with xylene (three times for 10 min each), dehydrated in a graded series of ethanol (100, 80 and 50%), and washed in running cold water for 10 min.

For antigen retrieval, slides were loaded in a slide holder and placed in a Nordicware microwave pressure cooker. 800 ml 10 mM citrate buffer pH 6.0 was added, and the pressure cooker was tightly closed and microwaved on high for 20 min. The slides were washed in cold running water for 10 min, soaked in 3% H₂O₂ in methanol for 10 min, and washed with warm running water for 10 min and with PBS for 3 min. Excess PBS was removed and a hydrophobic circle was drawn around the sample with a PAP pen (Sigma-Aldrich, St. Louis, MO). Anti-cleaved caspase 3 or anti-Ki-67 were diluted with Dako antibody diluent buffer (1:50, ~400 μl per slide), and incubated in a humidity chamber overnight at 4 °C. Slides were washed in PBS twice for 5 min each and ~4 drops of EnVision

+System-HRP labeled polymer (anti-rabbit) (Dako, Agilent Technologies, Santa Clara, CA) was added to each slide and incubated at room temperature for 30 min. Slides were washed again in PBS twice for 5 min each, and 4 drops of liquid diaminobenzidine + substrate chromogen (prepared according to manufacturer's instructions; Dako, Agilent Technologies) was added. The slides were developed for 5 min and rinsed under running cold water for 10 min. The slides were then soaked in 1% CuSO$_4$ for 5 min, rinsed briefly with running cold water, counterstained with haematoxylin for 60 s, and rinsed with running cold water. Next, slides were dipped in lithium carbonate three times, rinsed, and dehydrated in a graded series of ethanol. Coverslips were added, and the slides were examined with a Nikon Eclipse 80i microscope and photographed with a QImaging camera.

**Ethics approval**. We have complied with all relevant ethical regulations for human, animal testing and research. All relevant experiments in this study have received the appropriate ethical approval. The name of board and/or institution that approved the study protocol are described below.

Charles River Discovery Services North Carolina (CR Discovery Services) specifically complies with the recommendations of the Guide for Care and Use of Laboratory Animals with respect to restraint, husbandry, surgical procedures, feed and fluid regulation, and veterinary care. The animal care and use program at CR Discovery Services is accredited by the Association for Assessment and Accreditation of Laboratory Animal Care International, which assures compliance with accepted standards for the care and use of laboratory animals.

In Vivo Services at The Jackson Laboratory—Sacramento facility, an OLAW-assured and AAALAC-accredited organization conducted the DOHH2 mouse xenograft study. It was performed according to an Institutional Animal Care and Use Committee (IACUC)-approved protocol and in compliance with the Guide for the Care and Use of Laboratory Animals (National Research Council, 2011).

For the study using DLBCL lymphocytes, all procedures were approved and carried out in accordance with the guiding ethical principles of the Institutional Review Board of the Singapore General Hospital. Written informed consent was obtained for use of these samples for the specific research purpose only. The experimental protocol (#130812) was approved by the IACUC of the Biological Resource Center (BRC), A*STAR. All procedures involving human samples were approved by and performed in accordance with the ethics principles of the Sing Health Centralized Institutional Review Board. Written informed consent was obtained for use of these samples for the specific research purpose only. Our patients have given consent to use their tissue samples and associated demographical and clinical data in a de-identified format.

**Xenograft studies in mice**. DOHH2 xenograft study at Charles River's facility: Female severe combined immunodeficient mice (Fox Chase SCID®, C.B-17/Icr-Prkdcscid/IcrIcoCrl, Charles River) were 9 weeks old on Day 1 of the study and had a BW range of 17.8–22.9 g. The animals were fed ad libitum water (reverse osmosis, 1 ppm Cl) and NIH 31 Modified and Irradiated Lab Diet® consisting of 18.0% crude protein, 5.0% crude fat, and 5.0% crude fiber. On Day 1 of the study, animals were given a rehydration solution ad libitum in an effort to reduce dehydration during the dosing phase of the study. The rehydration solution consisted of 0.45% NaCl, 2.5% glucose, and 0.075% KCl in sterile water. The mice were housed on irradiated Enrich-o'cobs™ bedding in static microisolators on a 12-h light cycle at 20–22 °C (68–72 °F) and 40–60% humidity.

BL2 xenograft study at Jackson Laboratory: One hundred and five 6-week-old female NOD.CB17-Prkdc scid/J (NOD scid, Stock #001303) mice were transferred to the in vivo research laboratory in Sacramento, CA. The mice were ear notched for identification and housed in individually and positively ventilated polysulfone cages with HEPA filtered air at a density of 5 mice per cage. Initially cages were changed every two weeks. The animal room was lighted entirely with artificial fluorescent lighting, with a controlled 12 h light/dark cycle (6 a.m. to 6 p.m. light). The normal temperature and relative humidity ranges in the animal rooms were 20–26 °C and 30–70%, respectively. The animal rooms were set to have up to 15 air exchanges per hour. Filtered tap water, acidified to a pH of 2.5–3.0, and standard lab chow were provided ad libitum.

BL2 or DOHH-2 cells (1 × 10$^7$) and a cell suspension containing neoplastic DLBCL lymphocytes isolated from the pleural fluid of consented patient DLBCL3 were subcutaneously injected into the flank of immuno-compromised, female, NODscid mice at the Jackson Laboratory's, Charles River's, and Singapore General Hospital's facilities, respectively. After tumors formed, mice were divided into groups of approximately ten animals and given subcutaneous injections of vehicle daily, PCLX-001 daily at 10–60 mg/kg, or doxorubicin weekly at 3 mg/kg[70], as indicated in each figure. The dose volume was 10 mL/kg. At the end of the 2- to 3-week dosing period, mice were euthanized and three/group were necropsied. Mice that died or were euthanized early for humane reasons also were necropsied. In life, mice were monitored regularly and weighed daily, and tumors were measured with digital Vernier calipers (Mitutoyo) every other day. Tumor volume was calculated as length (mm) × width (mm)$^2$/2; length and width were the longest and shortest diameters, respectively. At euthanasia, at the end of the dosing period blood samples were taken for hematology analyses and clinical chemistry analyses that included AST and CK activities and bilirubin and creatinine concentrations (plus ALT activity and BUN concentration in the Jackson Laboratory study). At necropsy, samples of femur, both kidneys, liver, small intestine, and injection site

were collected and fixed. These were subsequently processed and examined by light microscopy for histopathologic findings. Also at necropsy, the tumors were removed and divided in two. One piece was fixed in 10% neutral buffered formalin for 24 h at room temperature and embedded in paraffin; the other was snap frozen for RNA and protein analysis. TGI for all xenograft experiments was calculated following the formula:

$$\text{TGI}(\%) = (V_{control} - V_{treated})/(V_{control} - V_{initial}) * 100.$$

**Patient derived xenograft mouse studies**

*(i) Patient sample.* Patient DLBCL3 had been treated for Stage I diffuse large B-cell lymphoma with cyclophosphamide, doxorubicin, vincristine, and prednisolone (CHOP), which resulted in complete remission (Supplementary Table 2). Patient DLBCL3 then presented to Singapore General Hospital 10 years subsequently with recurrent disease in the bone marrow and leptomeninges and pleural effusions. The patient received two courses of rituximab, ifosfamide, carboplatin, and etoposide and intrathecal methotrexate/cytarabine, followed by four courses of dexamethasone, cytarabine, and cisplatin and intrathecal methotrexate. The patient's tissue was harvested for PDX propagation at this time. The disease continued to progress, and the patient died a year later.

*(ii) Pathology.* Cytological examination of the pleural fluid showed discohesive lymphomatous population featuring large cells with vesicular chromatin and conspicuous nucleoli. Neoplastic cells expressed pan-B markers (PAX5, CD20, CD22, CD79a), with aberrant expression of CD5, strong expression of bcl2, and a high proliferation fraction (70–80%). Neoplastic lymphocytes had a nongerminal center phenotype (negative for CD10 and positive for bcl6, MUM1, FOXP1) but staining for c-Myc was low (20%). Interphase fluorescence in situ hybridization showed gains of *BCL2* and rearrangements of *BCL6* and *IGH*; normal patterns were seen for *C-MYC*. RNA in situ hybridization showed lack of *NMT2* expression.

*(iii) Xenograft construction and treatment.* The pleural fluid was collected in cold sterile 20% RPMI 1640 medium and neoplastic cells were isolated with Ficoll-Paque Plus (GE Healthcare) and re-suspended in RPMI 160 medium (Life Technologies) with 20% fetal bovine serum (Life Technologies, Carlsbad, CA). A representative part of the tumor sample was fixed in 10% neutral buffered formalin; the other part was used for xenotransplantation. The cell suspension was injected subcutaneously in the flank of 4–6-week-old NODscid mice. When the tumors reached a maximum of 1000 mm$^3$, the mice were sacrificed, tumors were harvested, and a necropsy was performed. Xenograft tumors were immediately frozen, fixed in formalin, and stored in 90% fetal bovine serum, and 10% dimethyl sulfoxide or placed in RPMI 1640 medium. This process was repeated to produce subsequent generations of patient-derived xenograft models (P2, P3, P4, …). To evaluate the maintenance of the morphology and main characteristics of the tumor of origin, formalin-fixed, paraffin-embedded tissue sections from patient tumor samples and xenografts of all established patient-derived xenograft models were stained with haematoxylin and eosin. These sections were also immunostained to measure the expression of various markers. A clinical pathologer reviewed all the slides. For the current study, tumor fragments (~50 mg, P4) were implanted subcutaneously in the flank of 4–6-week-old female NODscid mice and allowed to grow to 200–300 mm$^3$. The mice were then randomized into groups (n = 8 per group) and injected subcutaneously with vehicle (10 ml/kg); PCLX-001, 20 mg/kg daily for 21 days; or PCLX-001, 50 mg/kg daily for 18 days, with a 3-day break after 9 days. Tumor measurements and growth inhibition calculations were performed as described above.

For the DLBCL3 PDX study, NODscid mice were purchased from InVivos, Singapore and fed with standard laboratory diet and distilled water ad libitum. The animals were kept on a 12 h light/dark cycle at 22 ± 2 °C in BRC, A*STAR and maintained in accordance with the institutional guidelines.

**NMT activity assay**. NMT activity assay was described in Perinpanayagam et al.[33]. Briefly, cells were lysed and sonicated (10 s) in sucrose buffer (50 mM NaH$_2$PO$_4$, pH 7.4, and 0.25 M sucrose). Tumor samples were cut into small pieces, extracted by glass Dounce homogenization (12 full strokes) in sucrose buffer, and sonicated (10 s). The protein lysates were incubated with 0.1 mM of myristoylatable or non-myristoylatable decapeptide corresponding to the N-terminal sequence of p60-Src and 12 pM of [$^3$H]-myristoyl-CoA (PerkinElmer, Waltham, MA) in NMT assay buffer (0.26 M Tris-HCl pH 7.4, 3.25 mM EGTA, 2.92 mM EDTA and 29.25 mM 2-mercaptoethanol, 1% Triton X-100) in 25 μl reactions and incubated for 15 min at 30 °C. The reaction was terminated by spotting 15 μl of the reaction mixture onto a P81 phosphocellulose paper disc (Whatman, Maidstone, UK), washed and processed for scintillation counting.

**Statistical methods**. Data were analyzed using Prism 8 software (GraphPad, version 8.4.1) and generally expressed as mean ± s.e.m. Statistical significance was determined using Student t test or one-way ANOVA when applicable. Analysis of the significance of drug treatments on tumor volume was assessed by two-way ANOVA. P values higher than 0.05 were not considered statistically significant. ***$P \leq 0.001$, **$P \leq 0.01$, and *$P \leq 0.05$.

Statistical analysis of *NMT1* and *NMT2* expression: *NMT1* and *NMT2* mRNA expression data were extracted on March 26th 2020 from the Broad Institute CCLE database[54] (https://portals.broadinstitute.org/ccle) and contained the mRNA expression data for 1269 cancer cell lines. The RNAseq TPM gene expression data (Expression Public 20Q1) were analyzed for protein coding genes using RSEM and are presented as $Log_2$ transformed values using a pseudo-count of one (Supplementary Fig. 15).

**Reporting summary**. Further information on research design is available in the Nature Research Reporting Summary linked to this article.

## Data availability

Source data are provided with this paper. The data that support the findings of this study are available from the corresponding author upon reasonable request. Source data are provided with this paper.

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

## Acknowledgements
We thank Cheryl Santos for technical assistance. We thank Drs. Deanna Hockley, Larissa Vos, and Joseph Brandwein for their input and editing of the manuscript. We thank Dr. Marek Michalak (University of Alberta) for discussions and conceptual assistance with calcium measurements and Dr. J. Dillberger for analysis and interpretations of toxicology data. We sincerely thank Drs. Ong Choon Kiat and Dachuan Huang for providing us with the DLBCL3 PDX samples, which were used for the efficacy evaluation of PCLX-001. This work was supported by Alberta Cancer Foundation (ACF) grants 26362 and 26927, the Mary Agnes & Ivan Radostits ACF donor directed grant 26380 to LGB, by Eusera (www.eusera.com) and Pacylex Pharmaceuticals Inc. (www.pacylex.com) as well as an Alberta Innovates Translational Health Chair held by L.M.P.

## Author contributions
E.B., M.C.Y., M.A.P., A.I., J.M.G., K.M.V., and L.M.P. performed the in vitro and in silico experiments, provided results, prepared figures and assisted in the preparation of the manuscript. W-F.D. prepared and performed the pathology analysis of mouse tissue after necropsy. M.L., A.R., V.T., S.Y.T., S.T.L. identified the DLBCL patient for this study, established and performed the DLBCL patient derived xenograft and pathology analyses, as well as participated in the editing of the manuscript. K.D.R., D.W.G., and P.G.W. developed and provided PCLX-001, and, consulted on pharmacological aspects of in vitro and in vivo proof-of-concepts experiments. J.R.M. participated in the design of the in vivo experiments, and preparation of the manuscript. L.G.B. was involved in the design of the experiments, supervised the work, collected and integrated the data, wrote, and edited the final version of the manuscript.

## Competing interests
As co-founders of Pacylex Pharmaceuticals Inc. (www.pacylex.com), which owns the rights to patent applications PCT/2012/000696 and PCT/2013/050821, L.G.B., J.R.M., E.B., M.A.P., J.M.G., and M.Y. declare potential competing interests. To minimize these, cell line screens and animal xenograft studies were performed at arms length of the Pacylex co-founders. All the other authors declare no competing interests.
