## [Peer Review File · Nature Communications]

Reviewers' comments:

Reviewer #1 (Remarks to the Author); expert on lymphoma therapy:

The manuscript by Beauchamp and colleagues reports on the increased response of hematological malignant-derived cell lines, including B-cell lymphomas, to the pan-NMT inhibitor PCLX-001. They found that PCLX-001 treatment affected the global myristoylation of lymphoma cell proteins and also inhibited early BCR signaling by targeting Src kinases including LYN and BTK, leading to in vitro as well as in vivo dose-dependent cell killing. The work is of interest and provides new data on the use and mechanistic basis of action of this novel inhibitor inhibiting N-myristoylation as a potential anti-cancer agent, but I find a number of issues that need to be clarified:

Major issues

My major concern is whether (or up to what extent) the anti-tumor action of this compound observed in B-cell leukemias and lymphomas is purely related to the inhibition of BCR signaling. One intriguing issue relates to the fact that similarly to B-cell leukemias and lymphomas, anti-tumor responses to PCLX-001 are observed in 7 of 8 tested multiple myeloma cell lines (Fig 1A), a B-cell tumor that does not depend on BCR signaling for survival, and thereby a tumor that does not respond to BTK inhibitors in vitro nor in the clinic (i.e. to ibrutinib). In this line, very good responses in AML cell lines are observed (Fig 1B and Suppl Fig S2), another BCR-independent tumor (myeloid). These findings, together with the marked response observed in many (approx. from 1/3 to 1/4) tumor solid-derived cell lines, raise questions about the specificity of PCLX-001 action to kill the cells through BCR inhibition. While the effects of PCLX-001 treatment on BCR signaling in B-cell leukemia and lymphoma cell lines are well demonstrated, more experiments are required to evaluate possible non-BCR mechanisms leading to tumor cell death that could be consequence of targeting N-myristoylation.

Focusing on B-cell lymphomas, it is also noteworthy that PCLX-001 exhibits similar anti-tumor activity against activated B-cell (ABC) DLBCL cell lines (which clearly depend on BCR signaling for survival and respond to ibrutinib therapy) to that observed in germinal-center (GCB) B-cell lymphomas (GCB DLBCL and Burkitt lymphomas – tumors that do not show that high dependency on BCR signaling for surviving). Perhaps this could relate to the previously reported modification of HGAL protein by myristoylation in DLBCLs (a protein predominantly expressed by the GCB DLBCL subtype), which leads to its localization in cellular membrane rafts with consequences for BCR signaling through SYK interaction. This hypothesis might be explored.

One last suggestion that could strength the clinical and therapeutic importance of this work would be to evaluate whether primary patient tumor cells do respond to PCLX-001. This could easily be performed on peripheral blood samples from patients with CLL, a BCR-dependent B-cell malignancy where ibrutinib is approved for clinical use.

Reviewer #2 (Remarks to the Author); expert on BCR receptor:

The manuscript of Beauchamp et al on "Targeting N-myristoylation for therapy of B-cell lymphomas" describes the inhibition of B lymphoma growth and B cell signaling via the drug compound PCLX-001 that inhibits the activity of the N-myristoyl transferase NMT1 and NMT2 that mediated the N-terminal myristoylation of many proteins including the Src family kinase (SFK) Src and Lyn. Using a large panel of established leukemia lymphoma and myeloma cell lines as well as other tumor cell lines, the authors first show that, in a dose-dependent manner, PCLX-001 is most effective in inhibiting the growth of the B cell-derived tumors, in particular lymphomas. They then show that the treatment of lymphoma cell lines such as BL2 and IM9 with the most effective doses (0,1-1 μ M) of PCLX-001 inhibits the myristoylation and stability of Src and Lyn as well as their localization at the plasma membrane. The inhibition of constitutive signaling in diverse lymphoma cells by PCLX-001 is analyzed by Western blotting using anti-Lyn, anti-Src, anti-phospho-tyrosine and anti-phospho-SFK antibodies. These data show that PCLX-001 not only prevents

myristoylation, but also affects the stability of the SFKs as well as the phosphorylation of tyrosine substrate proteins. The authors then compare the efficiency of 0.1 to 1 μ M of PCLX-001 with similar doses of the anti-protein-tyrosine kinase inhibitor dasatinib and the BTK inhibitor ibrutinib and show that, in comparison to these established drugs, PCLX-001 is more effective in inhibiting tyrosine kinase activity of anti-IgM-stimulated B lymphoma lines. In addition, they show that PCLX-001 treated B lymphoma cell lines apparently have an increased ER stress response with an upregulated of BIP and PRAP1. In a dose response experiment involving many B cell tumor lines, PCLX-001 also compared favorable to that of dasatinib and ibrutinib in inhibiting the growth and viability of these cells. The authors then used NODscid mice as a xenograft model of human B lymphoma disease. These mice are first inoculated with cells of the DOHH2 or BL2 B lymphoma line or with human B cell tumor cells from a DLBCL3 patient. The mice are then treated with either PCLX-001 or, as positive control, with doxorubicin. The authors show that a dose between 20 and 60 mg/kg/day of PCLX-001 is inhibiting the growth of the human tumors and are more efficient than doxorubicin although in the latter case a dose of only 3mg/kg/day was used. In summary, this manuscript shows that PCLX-001 is a highly effective drug in inhibiting the growth and survival of B lymphoma cells.

Major points:

This manuscript clearly shows that the N-myristoylation inhibitor PCLX-001 is efficiently inhibiting the growth and survival of human B cell lines. The data showing the effect of this drug on normal B lymphocytes are limited. In their in vivo studies in supplementary Fig. S12 the authors show that mice treated with drugs have a weight loss, but little is known about the overall toxicity of this drug and its implication on the immune system. It thus would be important to include in this study also the effect of this drug on the normal immune system, for example, does PCLX-001 in the doses used in this study interfere with B and T cell development or immune system function? The author could cite in this context Rampoldi et al. *J Immunol* 2015; 195:4228-4243 and they should test how T cell and B cell responses are affected in PCLX-001 treated mice in comparison to control mice. For the latter they should monitor antibody titers after a vaccination procedure.

In Fig. 3 and Fig. 4 the authors show that PCLX-001 has an impact on the total phospho-tyrosine response. Another classical readout for B cell activation is calcium release and these data are missing from this manuscript. Therefore, the authors should monitor the anti-IgM as well as anti-IgD induced calcium release of B lymphoma cells treated previously with either dasatinib, ibrutinib or PCLX-001.

Minor points:

The authors discuss the effect of the PCLX-001 treatment mostly in context of the inhibition of Src family kinases, but this drug may have also other targets, which are more important for the growth inhibition and viability of B lymphoma cells. To address this, the authors could study whether or not PCLX-001 is still effective with B lymphoma cells treated with the SKI inhibitor PP2. Alternatively, the authors could employ Lyn and/or Src family kinase defective B cell lines and show whether in these cells the drug PCLX-001 is still effective. Such lines are available in several laboratories or can easily be generated via the CRISPR/Cas9 method.

Reviewer #3 (Remarks to the Author); expert on protein myristoylation:

The manuscript by Beauchamp et al. reports the discovery that hematological cancer cells are highly and more sensitive than other cancer cell lines to the compound called PCLX-001, an orally bioavailable derivative of the NMT inhibitor DDD85646. Furthermore, the authors show that PCLX-001 inhibits N-terminal protein myristoylation and induces apoptosis. Finally, the authors focus on the characterization of the effect of PCLX-001 on key BCR signaling proteins, in particular on different SFKs.

Overall, this is a very nice and clear manuscript and the first investigation that reports that specific cancer cells can be killed by an NMT inhibitor, at concentration that does not affect normal cells.

The finding that PCLX-001 is efficient in pre-clinical in vivo models of B-cell lymphoma is very exciting.

The experiments appear accurately performed and well presented but a number of points required clarification. Given the quality and scope of the manuscript, if the detailed comments below can be addressed, the paper is suitable for publication in Nature Comms.

Major comments

- 1) The authors nicely show that PCLX-001 is highly selective for hematological cancer cells. My major concern here is why? What about the protein level of NMT1 and 2 in the different cell lines? What about the permeability of PCLX-001 in the different cell lines?
- 2) Page 5, result section: where are the data of the third screen (China). I do not think that the journals accept now data not shown. This is the same for several other data reported as data not shown. The authors should show them or decide to omit them.
- 3) The authors used CellTiter Blue and Calcein assays to measure viable cells. Why are the results not comparable for some of the cell lines in Fig. 1E/F and Fig. S3A/B?
- 4) As the authors know, alkynyl myristate can be non-specifically incorporated on amino acids other than N-terminal glycines. To identify the YnMyr incorporation into N-terminal glycines and discriminate from the high level of non-myristoylation dependent background, they are few commonly accepted methods such as the use of NMT inhibitors or NaOH treatment. I'm surprise to see in Fig. 2A and even 2B practically no background dependent on non-myristoylation.
- 5) How can the authors explain that Tyr phosphorylation is strongly reduced in IM9 in Fig. S10A even if the SFK level is not in these cells (Fig.3A)? This is at least a different behavior than the BL2 cells.
- 6) Because the authors nicely show that non-myristoylated SFKs are part of an N-degron. It would have been nice to check the fate (protein level) of other well-known myristoylated proteins, at least for the well-known and abundant myristoylated proteasome subunit. However, I understand that this might be considered as perspective and maybe deserve to be investigated in the future.
- 7) The authors should clarify discrepancies between the right panel of Fig. 3B (24h) and data in Fig. S10A (i.e., the anti P-Ty (PY99) is strongly affected in BL2 in FigS10A (24h-treatment) but not in Fig. 3B).

Minor comments

- 1) Manuscripts on the Myristoylome (the set of proteins of a given proteome undergoing myristoylation) have been published recently. It would be fair that the authors update the literature, particularly in the introduction. For instance, as of now, it is thought that the human Myristoylome is composed by up to 600 proteins (Castrec et al. 2018) and not 200 as reported here.
- 2) Always in the introduction (page 4), when citing co and post-translational myristoylation, only a review for the latter one (review by the authors) is reported. Please, add also a review for the co-translational process.
- 3) Page 5, results section, please correct typo error "cell lines types" with cell line types.
- 4) I do not understand the difference between Fig1C and Fig. S1. They look the same to me. Hence, one of these should be removed.
- 5) Fig. S2, impossible to read the X axis legend. Please find another representation or split in two graphs.
- 6) Page 8, the authors should not report Fig. S8 and Fig. S9 together with Fig. 3B. The level of SFKs in PCLX-001-treated cell lines in Fig S8 and S9 are only in cells activated with IgM. Moreover, I would suggest adding to Fig.3B also the term Fig. 3A in the text, because more appropriate.
- 7) Fig. 2G, I suggest to improve the quality of the figure and separate the different membranes.
- 8) I do not agree with the sentence on page 9 "...supporting the established notion that non-myristoylated SFKs are not functional". This would be true only if the authors showed in vitro that the non Myristoylated proteins are not active. The protein is not functional because it is not longer localized in the appropriate compartment to phosphorylate the correct substrates and as shown later by the authors themselves because it undergoes degradation.....I would suggest reformulating the sentence and at least introducing the functional concept in vivo.

I. General comments:

Before I go into the details of our response to reviewers, I would like to state that this revised version includes further precisions of a conceptual nature throughout the manuscript since one of the reviewers seemed to have misunderstood the main point of the manuscript. Indeed, reviewer #1 seemed to think we meant to say our drug “PCLX-001 inhibits the B cell receptor in all cancer cells”, we do not. We mean to say PCLX-001 inhibits the myristoylation of numerous proteins and processes including the main survival pathway in B cell lymphomas, which is the B cell receptor signaling pathway (a fact well established in the literature), leading to the apoptotic death of malignant B cells.

Herein, I would like to add that we make three major claims in this manuscript:

1. To investigate whether NMTs are *bona fide* targets in cancer, we performed three robotics cancer line screens to evaluate their sensitivity of various cancer cells to a myristoylation inhibitor named PCLX-001. By doing so, we identified cancer cell types, including B cell lymphomas, selectively sensitive to NMT inhibition at a concentration much lower than that needed to kill normal cells. The implications of this discovery are of high clinical importance since this suggests a possible high therapeutic window for PCLX-001 that would allow the selective killing of cancer cells while sparing normal cells, potentially minimizing eventual side-effects in hematological cancer patients. Therefore, what really matters in the first part of the manuscript is that we are the first to demonstrate that hematological cancer cells are more sensitive to PCLX-001 than other types of cancer cells and normal cells. Importantly, this is an actionable discovery of high therapeutic potential in the clinic. To further strengthen this point, we have now added new data on the resistance of normal cells to the effect of PCLX-001 by characterizing its effects on normal HUVECs in Figure S5B in addition to the data we had already presented on normal blood cells (PBMCs and Lymphocytes; Fig. S5A).
2. We demonstrate how B cell lymphoma cells are more sensitive to myristoylation inhibition *in vitro* in the second part of the manuscript by showing a strong inhibition of the BCR signaling pathway, the main pro-survival pathway in B cell lymphoma. To further the BCR signaling inhibition point at the request of one of the reviewers, we are now showing that PCLX-001 also inhibits BCR-mediated calcium release, another hallmark of BCR signaling (NEW Fig. S12). This new experiment was carried out under the guidance of Dr. Marek Michalak, a world-renown calcium signaling expert. In addition, we clearly demonstrate that PCLX-001 is superior at inhibiting numerous aspects of BCR signaling *in vitro* and has the broadest spectrum of efficacy against malignant lymphoma cell lines than two clinically approved BCR downstream target inhibitors dasatinib (Sprycel) and ibrutinib (Imbruvica) with combined annual sales exceeding \$10B, this is a substantial finding. Please see our updated model in NEW Fig. 5 to summarize our new data. We also now show at one of the reviewer’s request that there was no synergy between ibrutinib or dasatinib and PCLX-001 suggesting PCLX-001 acts above the targets of these two clinically approved drugs (NEW Fig. S13).
3. We establish the first proof-of-concept that PCLX-001 potently kills B cell lymphoma cells not only *in vitro* but also *in vivo* using lymphoma cell lines and a drug resistant patient lymphoma tumour in xenograft models thereby supporting its ongoing pre-clinical development.

At the recommendations of reviewers, we have also added further details on the potential impact of PCLX-001 on various types of cancer cells, including those that do not express BCR, throughout the manuscript. In these other types of cancer cells other mechanisms could obviously be at play but may also involve processes altered by SFKs amongst other 600 myristoylated proteins known. We reiterate that we are well aware that PCLX-001 inhibits numerous pathways, did demonstrate this experimentally (now in Fig. 2A-C) and discussed these results accordingly in our original submission. While investigating the impact of our drug on every single proteins of the >600 myristoylated proteins in numerous cell lines is not feasible, we have taken the recommendations of reviewers#1 and #3 and assessed the impact of PCLX-001 on the myristoylated B cell protein HGAL, an enhancer of BCR signaling localized to lipid rafts (where the activated BCR receptor resides) and also the myristoylated Arf1 GTPase (a protein involved in vesicular transport pathway) to further illustrate that the effects of PCLX-001 are not restricted to myristoylated SFKs (see updates to Fig.4B,C). Also, at the request of reviewer#2, we have now evaluated the effects of PCLX-001, dasatinib and ibrutinib on BCR mediated calcium release in lymphoma cell line BL2 (NEW Figure S12 and see above comment).

In comment 11 (see response to reviewers), reviewer#3 states: *“The authors nicely show that PCLX-001 is highly selective for hematological cancer cells. My major concern here is why? What about the protein level of NMT1 and 2 in the different cell lines?”*

Finding out why hematological cell lines are more sensitive to PCLX-001 than other types of cancer cells may take years to demonstrate given the heterogenous nature of cancer cells in general and even within one cancer type. While the reasons for the increased sensitivity of hematological cancer cells to PCLX-001 are under active investigation, we have noted that NMT1 and NMT2 mRNA levels are altered in hematological cancer cells, are statistically lower and these data were presented at various scientific meetings.

To briefly address the request of reviewer#3, we could now include the bioinformatic data extracted from the CCLE database that show NMT1 and NMT2 expression levels are lowered in hematological cancer cells (see potential figure S15 on page 3 within) and expand on our originally discussed proposition that this may account for the sensitivity of hematological cancer cells to PCLX-001 in the discussion section then refer to this NEW Fig. S15 accordingly in the text (page 19-20, lines 427-441).

Unfortunately, this is only circumstantial evidence potentially linking the mRNA expression levels of *NMT1* and *NMT2* to the increased sensitivity of hematological cancer cells. Adding this type of comments in the discussion could raise further questions by the reviewers requiring experimental demonstration starting with whether or not mRNA expression correlate with protein levels in all these cell lines. In potential Fig. S15 attached, we show that while NMT2 levels are definitely lower in hematological cancer cells and especially so in B cell lymphoma, *NMT2* mRNA levels are very heterogenous (levels vary a lot) in these cancer cells and also in non-hematological cancer cell lines. Therefore, demonstrating a formal mechanistic link between the expression levels of *NMT1* and *NMT2* and the increased sensitivity of hematological cancer cells experimentally in a convincing manner would require both loss of function and gain of function (abrogation or addition) of both NMTs independently in numerous hematological cancer cell lines (likely >10 cell lines) and compare the results obtained to similarly modified non-hematological cancer lines (likely >10 cell lines).

Unfortunately, results from the above experiments would require a huge undertaking that would take years to perform and demand optimization of overexpression systems, transfection of siRNA or

shRNA or modification of numerous diploid cell lines with CRISPR Cas9 with the demonstration of the desired effects (abrogation or addition of NMTs) and would not significantly improve, further substantiate nor change the main conclusions of this manuscript. Furthermore, some unpublished preliminary experiments showed that re-expressing NMT2 in NMT2 deficient/low cell lines killed those cells, the re-expression of NMT2 in hematological cell might thus require the development of inducible expression methods to investigate its role in the resistance/sensitivity to PCLX-001 and adding to the already daunting task. While this is being investigated in our laboratory using various genomics and proteomics strategies, we think these lines of experimentation clearly belong in a subsequent manuscript. Again the key finding of this manuscript is that hematological cancer cells are exquisitely sensitive to myristoylation inhibition and that we can take advantage of this information pharmacologically for the first time to selectively kill cancer cells *in vitro* and *in vivo*.

Therefore, we suggest to simply leave the discussion as it basically was in the original manuscript and mention that the sensitivity of hematological cancer cells might be due to variations in NMT expression or variation in myristoylomes of hematological cancer cells and their reliance on these myristoylated proteins for survival (page 19, lines 427-441) and if required/requested by the reviewers and editor we could back up the NMT part of this statement by adding Fig.S15 in the supplementary material section of the final version of the manuscript and properly referring to it in the discussion. Importantly, including this supplementary figure might raise further questions by reviewers such as “How is NMT2 expression suppressed?” or “Why is NMT2 expression it suppressed?”, which are questions we are actively investigating and hopefully will be able to report in subsequent separate manuscripts. As stated above, this is the first demonstration that a potent NMT inhibitor can selectively kill hematological cancer cells while sparing normal cells *in vitro* and *in vivo*, we unfortunately cannot answer all conceivable questions in this first manuscript/publication.

As per the Nature Communications guidelines, we left the track changes function “on” before making the PDF to highlight the changes we made to the manuscript.

Having addressed the vast majority of the 25 reviewer concerns conceptually and experimentally, I hope you will agree with us that our manuscript is now suitable for publication in Nature Communications.

Potential Figure S15. *NMT* expression is decreased in hematological cancer cell lines. The average number of *NMT1* transcripts is larger than *NMT2* transcripts. However, *NMT2* transcript numbers (grey) show larger variations than *NMT1* transcript numbers (black) in cancer cell lines (A). *NMT2* mRNA expression is significantly lower in hematological cancer cell lines ($P < 0.0001$) in comparison to cell lines originating from other types of cancers (B). Expression of *NMT1* (C) is relatively constant across the 1269 cell lines investigated with a slight but significant decrease in expression in breast and leukemia cancer cell lines while *NMT2* expression (D) varies significantly amongst various cancers and also within a given cancer type. The data also illustrate that while the expression of *NMT2* is higher in cancer cell lines of CNS, kidney and fibroblast origins there is a selective and significant reduction of *NMT2* expression in hematological cancers such as leukemia, lymphoma and myeloma (D). Distribution of cancer types amongst the 100 cell lines with the lowest *NMT1* (E) and *NMT2* (F) expression levels shows enrichment in hematological cancers suggesting a key role for *NMT2* suppression in these types of cancers. *NMT1* expression is not increased in the 100 cells lines expressing the least *NMT2* as a possible compensatory mechanism (G). All data were extracted from 20Q1 PublicRNA-sequencing (Broad Institute, 1269 cell lines) and sorted in a selection of cancers. Box plots are showing 10-90 percentiles.

Potential Results: Fig. S15: Analysis of gene expression data from the Cancer Cell Line Encyclopedia¹ revealed first, that the *NMT1* number of transcripts is about eight times (2^3) the number of *NMT2* transcripts in all cell lines on average, and second, that there is a heterogeneous but significant reduction of *NMT2* expression in numerous hematological cancer cell lines in comparison to other types of cancer cell lines (Fig. S15A,B). Further analysis of the 100 cell lines expressing the lowest *NMT2* expression level revealed that 65 of those cell lines were of hematological origin (only 17 for *NMT1*, Fig. S15C,D). Interestingly, the low *NMT2* expression levels seen in lymphomas, leukemia and other cell lines were not compensated by an increase in *NMT1* expression (Fig. S15E). The number of *NMT1* transcripts were statistically lower than average in leukemia, breast cancer and colorectal cancer cell lines.

Potential Method : Fig, S15 Statistical analysis : *NMT1* and *NMT2* mRNA expression data were extracted on March 26th 2020 from the Broad Institute CCLE database¹ (<https://portals.broadinstitute.org/ccle>) and contained the mRNA expression data for 1269 cancer cell lines. The RNAseq TPM gene expression data (Expression Public 20Q1) were analysed for protein coding genes using RSEM and are presented as Log₂ transformed values using a pseudo-count of 1.

Potential Conclusion/Discussion: Fig. S15: Interestingly, these data show for the first time that in addition to be overexpressed in some cancers (aka the current dogma), *NMT* expression levels are actually lower in some cancers, many of which are of hematological origin. Altogether, these observations suggest a possible link between the reduction in the number of NMT enzyme targets in hematological cancer cells and the sensitivity of these cells to PCLX-001.

Reference:

1. Barretina J, *et al.* The Cancer Cell Line Encyclopedia enables predictive modelling of anticancer drug sensitivity. *Nature* **483**, 603-607 (2012).

II. Response to reviewers

Reviewer #1 (Remarks to the Author); expert on lymphoma therapy:

1. *“The manuscript by Beauchamp and colleagues reports on the increased response of hematological malignant-derived cell lines, including B-cell lymphomas, to the pan-NMT inhibitor PCLX-001. They found that PCLX-001 treatment affected the global myristoylation of lymphoma cell proteins and also **inhibited early BCR signaling by targeting Src kinases including LYN and BTK**, leading to in vitro as well as in vivo dose-dependent cell killing. The work is of interest and provides new data on the use and mechanistic basis of action of this novel inhibitor inhibiting N-myristoylation as a potential anti-cancer agent, but I find a number of issues that need to be clarified:*

From Reviewer#1 opening comment, we would like to specify that Bruton’s tyrosine kinase BTK is not a Src family kinase member, it is a Tec family tyrosine kinase that is not myristoylated. We added further precision on this in the revised manuscript lines 70-71 and included a reference. This may be of conceptual importance upon re-review.

Major issues:

2. *Reviewer#1’s major concern is whether (or up to what extent) the anti-tumor action of this compound observed in B-cell leukemias and lymphomas is purely related to the inhibition of BCR signaling.*

We do not claim that the anti-cancer action of PCLX-001 are solely or purely due to BCR inhibition in all B cell cancers aka B-cell leukemias or lymphomas as per the above reviewer comment. Rather, our data show and we mean to say that PCLX-001 inhibits the myristoylation of numerous proteins and processes including the main survival pathway in B cell lymphomas, which is the B cell receptor signaling pathway (a fact well established in the literature) leading to cell death. This role of pro-survival BCR signaling in B cell lymphomas is now further emphasized in lines 57-63 of the Introduction and we now added references #4-8 which include reviews highlighting this.

We also further discuss the role of BCR inhibition leading to cell death of B cell lymphoma cells on page 17 lines 378-387. In addition, we have made numerous subtle changes throughout the manuscript to further clarify this point. We have also removed the following sentence from the original manuscript *“Therefore our, PCLX-001 possibly represents a novel type of BCR signalling inhibitor that can specifically kill B lymphoma cells”* to further minimize possible confusion.

Again, that part of our manuscript only aims at elucidating the main mechanism of cell death induced by PCLX-001 treatment in B cell lymphoma and no other types of hematological or non-hematological malignant cells. It is clear from our data that PCLX-001 impacts the myristoylation numerous other proteins as originally/currently presented (Fig.2A-E) and discussed in the original and current versions of this manuscript. The discussion now extends to the abrogation of Arf1 and HGAL myristoylation by PCLX-001 (new Fig. 4B,C) in lines 389-398 and 412-420..

3. *Reviewer#1 had another intriguing issue relates to the fact that similarly to B-cell leukemias and lymphomas, anti-tumor responses to PCLX-001 are observed in 7 of 8 tested multiple myeloma cell lines (Fig 1A), a B-cell tumor that does not depend on BCR signaling for survival, and thereby a tumor that does not respond to BTK inhibitors in vitro nor in the clinic (i.e. to ibrutinib). In this line, very*

good responses in AML cell lines are observed (Fig 1B and Suppl Fig S2), another BCR-independent tumor (myeloid). These findings, together with the marked response observed in many (approx. from 1/3 to 1/4) tumor solid-derived cell lines, raise questions about the specificity of PCLX-001 action to kill the cells through BCR inhibition. While the effects of PCLX-001 treatment on BCR signaling in B-cell leukemia and lymphoma cell lines are well demonstrated, more experiments are required to evaluate possible non-BCR mechanisms leading to tumor cell death that could be consequence of targeting N-myristoylation.

Again, we do not claim that the anti-cancer action of PCLX-001 are solely or purely due to BCR inhibition in all cancers. Rather, our data show and we mean to say PCLX-001 inhibits the myristoylation of numerous proteins and processes including the main survival pathway in B cell lymphomas, which is the B cell receptor signaling pathway (a fact well established in the literature), leading to cell death (Please see comments above). We have made our best attempts to re-emphasize this throughout the manuscript.

Thus, there is no need to investigate all the non-BCR mechanisms of PCLX-001 in all other types of cancers. There are up to 600 known myristoylated proteins in cells and 1269 cancer cell lines in the Cancer Cell Line Encyclopedia (CCLE). We simply cannot investigate the effects of PCLX-001 in all hematological and non-hematological cancer cells. This is not feasible and is not the goal of this manuscript. Again, in this manuscript, we only aimed at: 1. evaluating the effects of a potent myristoylation inhibitor PCLX-001 on a large number of cancer cell lines (300) to potentially identify cancer cell lines (if any) that would be sensitive to PCLX-001, 2. establish our proof-of-concepts of the selective cell killing action of PCLX-001 *in vitro* and *in vivo* as well as 3. elucidating the mechanism of action of PCLX-001 in BL and DLBCL types of B cell lymphoma and no other types of hematological or non-hematological malignant cells. As for B cell lymphoma, as much as possible, we used 6 representative well characterized lymphoma cell lines and two normal immortalized B cell lines as controls to validate our observations.

4. Reviewer#1: *Focusing on B-cell lymphomas, it is also noteworthy that PCLX-001 exhibits similar anti-tumor activity against activated B-cell (ABC) DLBCL cell lines (which clearly depend on BCR signaling for survival and respond to ibrutinib therapy) to that observed in germinal-center (GCB) B-cell lymphomas (GCB DLBCL and Burkitt lymphomas – tumors that do not show that high dependency on BCR signaling for surviving). Perhaps this could relate to the previously reported modification of HGAL protein by myristoylation in DLBCLs (a protein predominantly expressed by the GCB DLBCL subtype), which leads to its localization in cellular membrane rafts with consequences for BCR signaling through SYK interaction. This hypothesis might be explored.*

While the reliance on BCR signals vary amongst the different sub-types of B cell lymphomas, our data clearly show PCLX-001 inhibits both the “chronic or tonic (antigen -independent)” as well as “antigen-dependent/receptor ligation” modes of BCR signaling leading to reductions of tyrosine phosphorylation of numerous proteins independent of the B cell lymphoma sub-type. The various types of BCR signaling modes are now introduced on lines 57-63 of the Introduction to add clarity.

To address another of the reviewer’s concerns, we now describe results on the effects of PCLX-001 on the myristoylated protein HGAL, which we introduce on lines 68-72. We now demonstrate that HGAL is extremely sensitive to PCLX-001 leading to its rapid degradation (NEW Fig. 4B). We discuss the rapid degradation of HGAL as likely occurring via the N-terminal Glycine specific N-Degron system recently identified by Timms et al (2019) in lines 401-420 (pages 18-19). This may cause the loss of

additional cell survival signals leading to cell death. In addition, to augment our data set on this topic, we also demonstrate that PCLX-001 treatment also promoted the degradation of the known myristoylated Arf1 GTPase protein (Fig. 4C). These results are described in lines 250-258 and their implications are discussed on page 18 lines 389-394. Overall, our data clearly show that the effects of PCLX-001 are not exclusive to SFKs but extend to other types of myristoylated proteins.

5. Reviewer#1: *One last suggestion that could strength the clinical and therapeutic importance of this work would be to evaluate whether primary patient tumor cells do respond to PCLX-001. This could easily be performed on peripheral blood samples from patients with CLL, a BCR-dependent B-cell malignancy where ibrutinib is approved for clinical use.*

Again, the goal of this manuscript is not about demonstrating that PCLX-001 inhibits the BCR signaling in all a BCR-dependent B-cell malignancies leading to cell death and that it does not inhibit the growth or viability of non-BCR dependent B cell malignancies. While comparing the effects of PCLX-001 vs the clinically approved ibrutinib on peripheral blood samples from patients with CLL is certainly of interest, we think this experiment clearly belongs in a different manuscript. This experiment is in the early planning phase actually. In addition, we do have a manuscript submitted in February 2019 that we are revising on the effects PCLX-001 in AML (now included. N.B. We anticipate the resubmission is >6-9months away since numerous animal experiments (xenografts) were requested by reviewers and we only received funding now to do them), again, this also clearly belongs in a different manuscript.

Reviewer #2 (Remarks to the Author); expert on BCR receptor:

6. Reviewer#2: *The manuscript of Beauchamp et al on "Targeting N-myristoylation for therapy of B-cell lymphomas" describes the inhibition of B lymphoma growth and B cell signaling via the drug compound PCLX-001 that inhibits the activity of the N-myristoyl transferase NMT1 and NMT2 that mediated the N-terminal myristoylation of many proteins including the Src family kinase (SFK) Src and Lyn. Using a large panel of established leukemia lymphoma and myeloma cell lines as well as other tumor cell lines, the authors first show that, in a dose-dependent manner, PCLX-001 is most effective in inhibiting the growth of the B cell-derived tumors, in particular lymphomas. They then show that the treatment of lymphoma cell lines such as BL2 and IM9 with the most effective doses (0,1-1 uM) of PCLX-001 inhibits the myristoylation and stability of Src and Lyn as well as their localization at the plasma membrane. The inhibition of constitutive signaling in diverse lymphoma cells by PCLX-001 is analyzed by Western blotting using anti-Lyn, anti-Src, anti-phospho-tyrosine and anti-phospho-SFK antibodies. These data show that PCLX-001 not only prevents myristoylation, but also affects the stability of the SFKs as well as the phosphorylation of tyrosine substrate proteins. The authors then compare the efficiency of 0.1 to 1 uM of PCLX-001 with similar doses of the anti-protein-tyrosine kinase inhibitor dasatinib and the BTK inhibitor ibrutinib and show that, in comparison to these established drugs, PCLX-001 is more effective in inhibiting tyrosine kinase activity of anti-IgM-stimulated B lymphoma lines. In addition, they show that PCLX-001 treated B lymphoma cell lines apparently have an increased ER stress response with an upregulated of BIP and PRAP1. In a dose response experiment involving many B cell tumor lines, PCLX-001 also compared favorable to that of dasatinib and ibrutinib in inhibiting the growth and viability of these cells. The authors then NODscid mice as a xenograft model of human B lymphoma disease. These mice are first inoculated with cells of the DOHH2 or BL2 B lymphoma line or with human B cell tumor cells from a DLBCL3 patient. The*

mice are then treated with either PCLX-001 or, as positive control, with doxorubicin. The authors show that a dose between 20 and 60 mg/kg/day of PCLX-001 is inhibiting the growth of the human tumors and are more efficient than doxorubicin although in the latter case a dose of only 3mg/kg/day was used. **In summary, this manuscript shows that PCLX-001 is a highly effective drug in inhibiting the growth and survival of B lymphoma cells.**

Excellent summary.

Major points:

7. Reviewer#2: *This manuscript clearly shows that the N-myristoylation inhibitor PCLX-001 is efficiently inhibiting the growth and survival of human B cell lines. The data showing the effect of this drug on normal B lymphocytes are limited. In their in vivo studies in supplementary Fig. S12 the authors show that mice treated with drugs have a weight loss, but little is known about the overall toxicity of this drug and its implication on the immune system. It thus would be important to include in this study also the effect of this drug on the normal immune system, for example, does PCLX-001 in the doses used in this study interfere with B and T cell development or immune system function? The author could cite in this context Rampoldi et al. J Immunol 2015; 195:4228-4243 and they should test how T cell and B cell responses are affected in PCLX-001 treated mice in comparison to control mice. For the latter they should monitor antibody titers after a vaccination procedure.*

We think the complete evaluation of the myristoylation inhibitor PCLX-001 on B and T cell development belongs in a different manuscript. We are well aware of the paper by Rampoldi et al. J Immunol 2015; 195:4228-4243 and to do a proper and complete analysis of PCLX-001 on B and T cell development would easily add another 4-5 figures to this manuscript and take significant amount of time. This would distract from this manuscript's main goals of evaluating the potential of NMT inhibition in cancers starting with lymphomas and showing an initial proof-of-concept backed up by a highly plausible mechanism of action for PCLX-001 tied in to BCR signaling inhibition, again, in B cell lymphoma.

At the request of the reviewer we now provide additional data on the lesser sensitivity of the normal human primary cells (HUVECs) to PCLX-001 in NEW Fig. S5B, include description of the results obtained on page 7, lines 151-153). Furthermore, to address the potential issues of toxicity raised by the reviewer, we have added more information on the toxicity/bio-safety of PCLX-001 in Table S2-S8, mention those results in lines 346-360 (see below for convenience) of this revised version of the manuscript and include a reference to a recent poster presented at the American Society for Hematology in 2019 (ref#55). The data presented in NEW tables S2-S8 demonstrate that no consistent drug related hematologic or biochemical abnormalities were identified in the xenograft studies to explain the key toxicity of dehydration and weight loss. As stated in the discussion and in toxicology summaries provided for each mouse xenograft study (see Table S2-S8), formal toxicology studies in other species are nearing completion.

“Mice tolerated PCLX-001 at efficacious dose levels.

Mice tolerated PCLX-001 at efficacious doses without specific end-organ toxicity. All mice treated with PCLX-001 survived the first xenograft study (Fig. 7A), while some mice treated with PCLX-001 at higher dose levels died in the other two studies (Fig. 7B,D). Neither the clinical pathology nor anatomic pathology evaluations identified the cause of death. Findings suggesting toxicity were seen in two studies. Of three mice bearing BL2 xenografts and given PCLX-001 at 50 mg/kg daily with a short treatment holiday, all had lower-than-normal neutrophil and lymphocyte counts at the end of the dosing period, and one also had lower-than-normal monocyte and platelet counts. In mice bearing DLBCL

lymphocyte xenografts and given PCLX-001 at 20, 50, or 60 mg/kg daily, signs of ill health (e.g., rough and scruffy coats, piloerection) were noted in most mice at all dose levels, and dehydration and weight loss were noted at 50 and 60 mg/kg daily.”

8. Reviewer#2: *In “the original” Fig. 3 and Fig. 4 the authors show that PCLX-001 has an impact on the total phospho-tyrosine response. Another classical readout for B cell activation is calcium release and these data are missing from this manuscript. Therefore, the authors should monitor the anti-IgM as well as anti-IgD induced calcium release of B lymphoma cells treated previously with either dasatinib, ibrutinib or PCLX-001.*

At the request of reviewer#2 and for added consistency with figures 3 and 4, we are now showing the inhibitory effect of PCLX-001, dasatinib and ibrutinib on BCR mediated calcium release in BL2 cells in NEW Fig. S12. Those new results are now introduced on lines 73-75, described on lines 276-286 and discussed on lines 379-387.

Minor points:

9. Reviewer#2: *The authors discuss the effect of the PCLX-001 treatment mostly in context of the inhibition of Src family kinases, but this drug may have also other targets, which are more important for the growth inhibition and viability of B lymphoma cells. To address this, the authors could study whether or not PCLX-001 is still effective with B lymphoma cells treated with the SKF inhibitor PP2. Alternatively, the authors could employ Lyn and/or Src family kinase defective B cell lines and show whether in these cells the drug PCLX-001 is still effective. Such lines are available in several laboratories or can easily be generated via the CRISPR/Cas9 method.*

We have performed the combination of PCLX-001 with either the SFK inhibitor dasatinib or the BTK inhibitor ibrutinib as requested by the reviewer and monitored the effect of combining these drugs together on malignant B cell viability. To do so, we combined either dasatinib or ibrutinib at concentrations of 0.1 and 1.0 μ M to PCLX-001 at 0.01, 0.1 and 1.0 μ M. Those treatments did not result in any additional cell growth rate inhibition or any further decrease in viability suggesting that PCLX-001 effects are broader than those of dasatinib and ibrutinib or that it acts upstream of dasatinib and ibrutinib targets (see NEW Fig S13).

Since there are 9 different SFKs in cells with different sub-type expression patterns and that these SFKs all have somewhat redundant activities, the proper testing of reviewer#2 hypothesis would require the CRISPR/Cas9 knock out of numerous combinations of SFKs (not only Src and Lyn) in numerous cell types to be conclusive. This experiment would require an enormous amount of work, is impractical and likely to give inconclusive results (e.g. if there are still some SFKs (e.g Lck, Fyn, Hck etc...) that can compensate for the activity of the lost Lyn and Src double KO as per example). We do not think this experiment is needed to validate the conclusions drawn in our manuscript.

Reviewer #3 (Remarks to the Author); expert on protein myristoylation:

10. Reviewer#3: *The manuscript by Beauchamp et al. reports the discovery that hematological cancer cells are highly and more sensitive than other cancer cell lines to the compound called PCLX-001, an orally bioavailable derivative of the NMT inhibitor DDD85646. Furthermore, the authors*

show that PCLX-001 inhibits N-terminal protein myristoylation and induces apoptosis. Finally, the authors focus on the characterization of the effect of PCLX-001 on key BCR signaling proteins, in particular on different SFKs.

Overall, this is a very nice and clear manuscript and the first investigation that reports that specific cancer cells can be killed by an NMT inhibitor, at concentration that does not affect normal cells. The finding that PCLX-001 is efficient in pre-clinical *in vivo* models of B-cell lymphoma is very exciting.

The experiments appear accurately performed and well presented but a number of points required clarification. **Given the quality and scope of the manuscript, if the detailed comments below can be addressed, the paper is suitable for publication in Nature Comms.**

Excellent summary and **we agree with reviewer#3** that having performed the large majority of experiments requested by the three reviewers that **this manuscript deserves to be published in Nature Communications.**

Major comments

11. Reviewer#3: The authors nicely show that PCLX-001 is highly selective for hematological cancer cells. My major concern here is *why*? What about the protein level of NMT1 and 2 in the different cell lines? What about the permeability of PCLX-001 in the different cell lines?

We do not know exactly **why** hematological cancers as a whole are more sensitive to PCLX-001 than other types of cancer cells. As stated, in our opening comments, it may take years to find this out. We had briefly discussed this issue in the original manuscript by mentioning that either NMT levels in cells or the levels of myristoylated proteins critical for hematological cancer cell survival or both are altered in these cells as an explanation. Again, as mentioned in our opening letter, we could leave this statement pretty much as is (see Discussion, page 19, lines 427-441) or back up part of that statement by providing bio-informatic data (NEW potential Fig. S15) from the CCLE database showing that *NMT* levels are lower in hematological cancer cell lines and may represent a plausible reason for their increased sensitivity to PCLX-001. Interestingly, these data show for the first time, that in addition to be overexpressed in some cancers (aka the current dogma), *NMT* expression levels are actually lower in other cancers, many of which are of hematological origin.

Given the heterogenous and large variations of *NMT1* and *NMT2* expression in various cancer cell line types and even within each type of cancer cell lines, testing the hypothesis that PCLX-001 sensitivity relates to the expression levels of NMT1 and NMT2 in cancer cells would be a daunting task requiring ablating or overexpressing either NMT in a large number of cell lines to be able to draw a reliable conclusion. This is nearly impossible to do and clearly belongs outside the scope of this manuscript.

While we do not know **why** hematological cells are more sensitive to PCLX-001, the critical information provided in our manuscript is that we show that hematological cancer cells (especially lymphoma cells) are more sensitive to PCLX-001 than normal cells and that we can take pharmacological advantage of that fact to selectively kill these cancer cells *in vitro* and *in vivo* with the hope of eventually applying our findings to treating lymphoma patients.

12. Reviewer#3: *Page 5, result section: where are the data of the third screen (China). I do not think that the journals accept now data not shown. This is the same for several other data reported as data not shown. The authors should show them or decide to omit them.*

Unfortunately, reviewer#3 may have missed these data since they were in the original manuscript as Figure S2 and still are in the current version. Of note, this screen was performed on a different platform and the data collected as reported to us by ChemPartners did not allow the calculation of the % of Growth Inhibition since the original number of cells plated is missing from their data set. We are thus only showing the PCLX-001 IC50 of their 131 cancer cell lines tested.

13. Reviewer#3: *The authors used CellTiter Blue and Calcein assays to measure viable cells. Why are the results not comparable for some of the cell lines in Fig. 1E/F and Fig. S3A/B?*

We are now providing a detailed explanation of what is measured by both assays on page 6 (line 133-138). 1) CellTiter Blue assay results are dependent on both cell proliferation and cell viability. This assay evaluates the total number of viable cells, 2) Calcein assay results are independent of proliferation rate and only measure the percentage of viable cells in a cell population and, 3) the proliferation assay simply counts the total number of cells over time independently of their viability.

For example, 1000 cells at 50% viability (500 viable cells) would give a higher signal with the CellTiter Blue than 500 cells at 80% viability (400 viable cells). However, the Calcein assay gives us the information that the second set is more resistant to PCLX-001. The combination of the 3 assays allows us to fully understand the effect of PCLX-001 on both proliferation and viability for a particular cell line.

14. Reviewer#3: *As the authors know, alkynyl myristate can be non-specifically incorporated on amino acids other than N-terminal glycines. To identify the YnMyr incorporation into N-terminal glycines and discriminate from the high level of non-myristoylation dependent background, they are few commonly accepted methods such as the use of NMT inhibitors or NaOH treatment. I'm surprise to see in Fig. 2A and even 2B practically no background dependent on non-myristoylation.*

Alkynyl fatty acids have been used in our laboratory for many years and we have optimised the incubation conditions (time and concentration) to maximize the specificity of incorporation. Our methodology was fully described in Yap *et al.* (Journal of Lipid Research, 2010, reference #48). To reduce the background possibly due to incorporation of the myristate analog ω -alkynyl myristic at palmitoylatable sites, we only incubate the cells for 30 minutes with the ω -alkynyl myristic as described in the Method section. Therefore, because of the extent of optimization and use of this method over the years, we do not need to validate our results by soaking the membranes in alkaline solution to remove the ω -alkynyl myristic possibly linked to proteins via alkali sensitive thioester or oxyester bonds.

When we needed to measure palmitoylation, we incubated the cells for 4 hours with ω -alkynyl palmitic as described within the Method section. Despite these optimized conditions, some background can still be observed around the 75kDa MW due to the presence of a naturally biotinylated protein, likely a mitochondrial decarboxylase.

15. Reviewer#3: *How can the authors explain that Tyr phosphorylation is strongly reduced in IM9 in*

Fig. S10A even if the SFK level is not in these cells (Fig.3A)? This is at least a different behavior than the BL2 cells.

We think reviewer#3 means Src SFK? If this is the case, while the levels of Src SFK do not change as much in IM-9 cells as in other malignant cell lines treated with PCLX-001 they still do change and go down (please also see Fig. 2F and its quantification in Fig. S7), the levels of Lyn SFK (the main SFK associated with BCR) readily go down in a time and concentration manner in IM9 cells (Figure, 3A) and it might be the case for some of the other 7 SFKs in IM9 cells. Importantly we show the levels of both Src SFK and Lyn SFK to go down in 7 of 7 other cell lines tested in Fig. 3A. Furthermore, we show the levels of 2 other SFKs also go down in BL2 cells, namely Hck and Lck in Fig. 3B. Importantly, the anti-P-SFK antibody measures the conserved active site phosphorylation of ALL SFKs. Of critical importance, we show the levels of the anti-P-SFK goes down drastically in ALL B cell lymphoma cells treated with PCLX-001 tested in this manuscript.

Of note, the basal (antigen or ligand independent) level of phospho-tyrosine proteins detected by the PY99 antibody is drastically lower in normal IM9 B cells than the phosphor-tyrosine levels observed for the other malignant cell lines (the blots presented in Fig.S8A were all loaded with the same amount of protein and were all exposed for the same amount of time).

16. Reviewer#3: *Because the authors nicely show that non-myristoylated SFKs are part of an N-degron. It would have been nice to check the fate (protein level) of other well-known myristoylated proteins, at least for the well-known and abundant myristoylated proteasome subunit. However, I understand that this might be considered as perspective and maybe deserve to be investigated in the future.*

We agree with reviewer#3 and we have now provided two more examples of myristoylated proteins affected by PCLX-001: HGAL and Arf1. The effects of PCLX-001 on HGAL were also requested by reviewer#1. We now present the new data showing treatment of cells with PCLX-001 reduces the level of HGAL and Arf1 in a concentration dependent manner in Fig. 4B,C. Please note the exquisite sensitivity of the HGAL BCR signaling enhancer protein to PCLX-001 treatment. The analysis of the impact on PCLX-001 on other myristoylated proteins (there are up to 600 of them) belongs in a different manuscript.

17. Reviewer#3: *The authors should clarify discrepancies between the right panel of Fig. 3B (24h) and data in Fig. S10A (i.e., the anti P-Ty (PY99) is strongly affected in BL2 in FigS10A (24h-treatment) but not in Fig. 3B).*

These are not discrepancies. The difference between Fig. 3 and S10A is due to the fact that we are looking at phospho-tyrosine (P-Y) levels using anti-PY99 antibody in BL2 cells after IgM activation (mimicking antigen dependent) of BCR signaling in Fig. 3A while we are looking at basal phospho-tyrosine levels corresponding to tonic or chronic (antigen independent) BCR signaling for numerous cell lines in figure S10A. The two signaling conditions are compared side-by-side in Fig. S10B for BL2 cells.

We are now providing additional explanations on the diverse modes of BCR signaling in the introduction lines 57-63 to further clarify the differences.

Minor comments

18. Reviewer#3 *Manuscripts on the Myristoylome (the set of proteins of a given proteome undergoing myristoylation) have been published recently. It would be fair that the authors update the literature, particularly in the introduction. For instance, as of now, it is thought that the human Myristoylome is composed by up to 600 proteins (Castrec et al. 2018) and not 200 as reported here.*

We edited the introduction to account for this recent publication. The new reference by Castrec *et al* (#26 in the manuscript) has now been inserted on page 4 line 89.

19. Reviewer#3. *Always in the introduction (page 4), when citing co and post-translational myristoylation, only a review for the latter one (review by the authors) is reported. Please, add also a review for the co-translational process.*

While our review also covered the topic of co-translational myristoylation despite its title, we have now edited the introduction to include the review by the Tate group, which covers myristoylation more broadly at the recommendation of the reviewer (reference#25 in the manuscript).

20. Reviewer#3 *Page 5, results section, please correct typo error “cell lines types” with cell line types.*

We have now corrected this typographical error.

21. Reviewer#3 *I do not understand the difference between Fig1C and Fig. S1. They look the same to me. Hence, one of these should be removed.*

Fig. S1 represent the results for the total 169 cell lines studied in Fig.1 (e.g. the combination of the results observed in Fig. 1B (68 cell lines) to those in Fig. 1C (101 cell lines)).

22. Reviewer#3 *Fig. S2, impossible to read the X axis legend. Please find another representation or split in two graphs.*

We have now improved the legibility of Fig. S2.

23. Reviewer#3 *Page 8, the authors should not report Fig. S8 and Fig. S9 together with Fig. 3B. The level of SFKs in PCLX-001-treated cell lines in Fig S8 and S9 are only in cells activated with IgM. Moreover, I would suggest adding to Fig.3B also the term Fig. 3A in the text, because more appropriate.*

We agree with this comment from reviewer#3. While results in Fig. 3B are also from cells activated with anti-IgM we have now used different sentences to better separate the description of results from fig. S8 and S9 from those of Fig. 3B. lines 232-248.

24. Reviewer#3 *Fig. 2G, I suggest to improve the quality of the figure and separate the different membranes.*

We have now improved the cosmetic appearance of Fig 2G and separated it in three pieces as per the reviewer recommendation.

25. Reviewer#3 *I do not agree with the sentence on page 9 "...supporting the established notion that non-myristoylated SFKs are not functional". This would be true only if the authors showed in vitro that the non Myristoylated proteins are not active. The protein is not functional because it is not longer localized in the appropriate compartment to phosphorylate the correct substrates and as shown later by the authors themselves because it undergoes degradation.....I would suggest reformulating the sentence and at least introducing the functional concept in vivo.*

We have edited the sentence accordingly (page 10, lines 225-226).

REVIEWERS' COMMENTS:

Reviewer #1 (Remarks to the Author):

The authors have made extensive efforts to improve the manuscript, which include addressing most of my concerns in the original review.

Reviewer #2 (Remarks to the Author):

The authors have now improved their manuscript on the therapy of B-cell lymphomas by with the myristoylation inhibitor PCLX-001 according to some of my suggestions. They now describe that this inhibitor is not only targeting Src family kinases, but also many other myristoylated proteins including HGAL and rab proteins. Although they have not taken up my suggestion to test the combination of the Src-family kinase inhibitor PP2 with PCLX-001 they showed that downstream inhibitors of BCR signaling such as dasatinib and ibrutinib do not improve or alter the reactivity of PCLX-001. I still find it disappointing that the authors do not provide some more data on the effect of the drug PCLX-001 on normal lymphocyte function in their experimental mice. I think that such data will be anyhow required for the approval of the drug and maybe this information will be given by the authors in another publication. In summary, I do not have any further suggestions for the improvement of the manuscript and suggest its publication as such.

Michael Reth

Reviewer #3 (Remarks to the Author):

The manuscript by Beauchamp et al. reports the discovery that hematological cancer cells are highly and more sensitive than other cancer cell lines to the compound called PCLX-001, an orally bioavailable derivative of the NMT inhibitor DDD85646.

Concerning my first query, I would suggest simplifying FigS15 to make as clear as possible the home-take-message "NMT expression levels are lower in some cancers including that of hematological origin". For instance, I wonder if Figure S15E and S15F are necessary. Certainly, this analysis and the low level of NMT cannot alone explain the selectivity of the inhibitor but, as the authors say, it will once again help to underline the fact that in some tumors the NMT level has not increased, on the contrary.

Overall, I am satisfied that the authors have addressed most of my concerns. Not being an expert on BCR receptors or lymphoma therapy, I will leave my colleagues to judge their specific queries.

Reviewer #4 (Remarks to the Author):

The manuscript by Beauchamp et al entitled "Targeting N-myristoylation for therapy of B-cell lymphomas" describes a characterization of the efficacy of PCLX-001 for tumor growth inhibition. The authors show some evidence of mechanism via degradation of BCR signaling components due to loss of myristoylation and deficient cellular localization. PCLX-100 inhibits myristoylation of Src and Lyn and prevents proper localization at the plasma membrane.

The manuscript is well written and fairly comprehensive in scope. For example, 300 cancer cell lines were screened to identify the most sensitive therapeutic target tumors. The results show that B Cell derived tumor cells were the most susceptible to PCLX-100 and therefore suggest that lymphomas would be the likely therapeutic target of PCLX-001, and potentially other NMT

inhibitors. As a comparator, the IM-9 cell line (described by the authors as benign) was mainly used to assess tumor cell selection. In addition, some comparative work was done in isolated PBMCs. PCLX-100 killed B cell tumor cells in vitro more effectively than dasatinib or imatinib at the same concentrations.

The authors also used 3 xenograft tumor models (one of them a PDX) in SCID mice and treating with PCLX-100 and Doxorubicin as an approved comparator. PCLX-100 showed efficacy in all to some degree at tolerated doses.

Major Comments:

STD10 values were exceeded in SCID mice (as evidenced by the deaths within the 7 animal group) at 50 mg/kg in both BL2- and DLBCL3-derived tumors models at fully efficacious doses. At tolerated doses of 20 mg/kg in BL-2 and DLBCL3 appear to be only marginally efficacious, with $\sim \leq 66\%$ tumor size reduction (Fig 7). This should be addressed by the authors in the discussion. Where systemic exposures collected in the xenograph studies for comparison to the in vitro concentrations that were effective at killing tumor cells?

The authors mention that there were no specific end organ toxicities in mice that tolerated PCLX-100 at efficacious doses. Which tissues were examined in animals found dead – was this a comprehensive tissue list or only lymphoid tissues? What endpoints were evaluated by clinical pathology?

A comparison to a greater variety of rapidly dividing primary cultures (ie GI epithelium, hematopoietic, skin cells) typical of oncology target tissues would give much greater confidence in the safety of PCLX-100. In fact, PBMCs (normal cells) and BL2 cell IC50's appear similar (~ 0.03 - 0.1 μM) based on Figs 1 and S5. This should be stated in the absence of additional cytotoxicity assays in more normal cell types. Based on the benefit/risk for this therapeutic indication, it is acceptable and not unusual that some normal tissues are effected at efficacious doses.

Minor Comments:

The authors should be commended for conducting an extensive screen of kinases. Cross species pharmacology is not confirmed with this platform containing human kinases, and it should be described as a "human" kinase screen. Was the screening panel used in this study the preconfigured Eurofins "scanMAX" KINOMEScan or a custom panel of 468 kinases?

The authors use the term "robotic" repeatedly throughout the manuscript (12 times). It is not necessary to repeat this detail throughout the manuscript.

Decreases in the expression in non-myristoylated proteins may be due to cell death, which often causes a general decrease in gene/protein expression (although increases in some specific proteins may also be associated with cell death depending on mechanism/compensatory pathways). In fact, it appears that these decreases coincide with cytotoxic concentrations of PCLX-100. The authors should discuss/address this possibility.

What was the rationale for using SQ route of administration for xenograft studies? The text states that PCLX-100 is an orally bioavailable small molecule, which suggests clinical dosing will be by the oral route?

(Lines 145-149) Why use 100 nM vs 0.1 μM ? I suggest using consistent units throughout.

Resistant PBMCs tolerated up to 10 μM (Lines 149-151). What about resistant tumor cells (ie BL2)? Why point out this observation without a comparison to resistant tumor cells?

(Lines 290-293) Comparison's to dasatinib and Ibrutinib are really comparing potency. Due to tolerability considerations, it may be possible that these other kinase inhibitors are better tolerated and can be dosed at higher levels than PCLX-100 leading to more efficacy. The important consideration is efficacy compared to toxicity. The authors should describe that this is at the same concentrations and not make general statements about more effectively killing tumor cells in general.

(Line 340) – "in" can be removed before apoptosis (may be an orphan from previous revisions)

(Line 372-376) – Correct for accuracy PBMC's (normal cells) similar in sensitivity, whereas IM9 are not "normal" cells, but an immortalized cell line.

(Line 594) – Should be binding vs "biding"

(Line 733) – Why did a clinical pathologist review the slides versus an anatomical pathologist?

REVIEWERS' COMMENTS:

Reviewer #1 (Remarks to the Author):

The authors have made extensive efforts to improve the manuscript, which include addressing most of my concerns in the original review.

Agreed.

Reviewer #2 (Remarks to the Author):

The authors have now improved their manuscript on the therapy of B-cell lymphomas by with the myristoylation inhibitor PCLX-001 according to some of my suggestions. They now describe that this inhibitor is not only targeting Src family kinases, but also many other myristoylated proteins including HGAL and rab proteins. Although they have not taken up my suggestion to test the combination of the Src-family kinase inhibitor PP2 with PCLX-001 they showed that downstream inhibitors of BCR signaling such as dasatinib and ibrutinib do not improve or alter the reactivity of PCLX-001.

Rather than using PP2 as a Src inhibitor, we used the clinically approved drug dasatinib and provided a reference that it is a *bona fide* Src tyrosine kinase inhibitor within the body of

the text. We still think our experiment was superior to the one recommended by reviewer#2.

I still find it disappointing that the authors do not provide some more data on the effect of the drug PCLX-001 on normal lymphocyte function in their experimental mice. I think that such data will be anyhow required for the approval of the drug and maybe this information will be given by the authors in another publication. In summary, I do not have any further suggestions for the improvement of the manuscript and suggest its publication as such.

Reviewer #2 should not be disappointed, these data belong in a second manuscript but is overall correct, we have now completed an extensive series of pre-clinical biosafety evaluation of PCLX-001 in rats and dogs as part of our imminent application for a human Phase I clinical trial application (CTA) (CTA in Canada in equivalent to IND in the US). The vast amount of new toxicology results will also be reported elsewhere when available. We are still waiting for the final GMP biosafety study report.

Reviewer #3 (Remarks to the Author):

The manuscript by Beauchamp et al. reports the discovery that hematological cancer cells are highly and more sensitive than other cancer cell lines to the compound called PCLX-001, an orally bioavailable derivative of the NMT inhibitor DDD85646.

Concerning my first query, I would suggest simplifying FigS15 to make as clear as possible the home-take-message "NMT expression levels are lower in some cancers including that of hematological origin". For instance, I wonder if Figure S15E and S15F are necessary. Certainly, this analysis and the low level of NMT cannot alone explain the selectivity of the inhibitor but, as the authors say, it will once again help to underline the fact that in some tumors the NMT level has not increased, on the contrary.

We agree with reviewer#3 and have edited supplementary figure 15 accordingly and have removed supplementary figure panels 15E and 15F to improve clarity. To describe the new requested data, we added a paragraph on page 14, lines 421-444 and commented on the new statistics in the Method section.

Overall, I am satisfied that the authors have addressed most of my concerns. Not being an expert on BCR receptors or lymphoma therapy, I will leave my colleagues to judge their specific queries.

We agree.

Reviewer #4 (Remarks to the Author):

Comment 1: The manuscript by Beauchamp et al entitled "Targeting N-myristoylation for therapy of B-cell lymphomas" describes a characterization of the efficacy of PCLX-001 for tumor growth inhibition. The authors show some evidence of mechanism via degradation of BCR signaling components due to loss of myristoylation and deficient cellular localization. PCLX-100 inhibits myristoylation of Src and Lyn and prevents proper localization at the plasma membrane.

The manuscript is well written and fairly comprehensive in scope. For example, 300 cancer cell lines were screened to identify the most sensitive therapeutic target tumors. The results show that B Cell

derived tumor cells were the most susceptible to PCLX-100 and therefore suggest that lymphomas would be the likely therapeutic target of PCLX-001, and potentially other NMT inhibitors. As a comparator, the IM-9 cell line (described by the authors as benign) was mainly used to assess tumor cell selection. In addition, some comparative work was done in isolated PBMCs.

Technically, we used IM-9, VDS, PBMCs and lymphocytes as immortalized non-malignant (2) cells or normal cells (2) as controls respectively to assess the therapeutic window of PCLX-001 and therefore evaluate its ability to selectively/preferentially kill malignant cells at concentrations that would spare immortalized or normal cells. In the revised manuscript resubmitted in May 2020, we had also added the effect of PCLX-001 on HUVECs (now line 189) (Supplementary Fig. 5B).

Comment 2: PCLX-100 killed B cell tumor cells in vitro more effectively than dasatinib or imatinib at the same concentrations.

The authors also used 3 xenograft tumor models (one of them a PDX) in SCID mice and treating with PCLX-100 and Doxorubicin as an approved comparator. PCLX-100 showed efficacy in all to some degree at tolerated doses.

Correct.

Major Comments:

Comment 3: STD10 values were exceeded in SCID mice (as evidenced by the deaths within the 7 animal group) at 50 mg/kg in both BL2- and DLBCL3-derived tumors models at fully efficacious doses. At tolerated doses of 20 mg/kg in BL-2 and DLBCL3 appear to be only marginally efficacious, with $\sim \leq 66\%$ tumor size reduction (Fig 7). This should be addressed by the authors in the discussion.

We have described these results rather extensively in the result section lines 448-515 for the effects of PCLX-001 treatments on tumours and lines 517-535 for the effects of the compound on the mice (toxicology) and showed that most effects on mouse body weights were readily reversible and likely due to dehydration.

To take the reviewer#4 comment into consideration we have added the following statement linking PCLX-001 dosing to efficacy in the discussion lines 643-650:

“While PCLX-001 was only marginally efficacious at the tolerated dose of 20mpk [$\sim 66\%$ tumour reduction (Fig.7)], we show that it effectively inhibits tumor cell growth *in vivo* resulting in either major or complete regression of disease in three human lymphoma xenograft models at the relatively well tolerated 50mpk efficacious dose, including complete response in a lymphoma refractory to CHOP, Rituximab and other salvage therapies.”

Comment 4: Where systemic exposures collected in the xenograph studies for comparison to the in vitro concentrations that were effective at killing tumor cells?

We haven't done these studies in mice but they were done in rats and dogs as part of our pre-clinical biosafety studies *en route* to human Phase I studies. These results will be part of a subsequent manuscript.

Comment 5: The authors mention that there were no specific end organ toxicities in mice that tolerated PCLX-100 at efficacious doses. Which tissues were examined in animals found dead – was this a comprehensive tissue list or only lymphoid tissues? What endpoints were evaluated by clinical pathology?

Animals found dead were necropsied and examined for grossly visible findings by our clinical pathologist and co-author Dr. Wei-feng Dong (somehow Dr. Dong had disappeared from the author list in the revision cycles but was added back on line 4), but the only tissue samples collected and examined microscopically were those listed in the Materials and Methods: kidneys, liver, small intestine, femur, and injection site (see lines 990-992). The clinical pathology endpoints included in hematology analyses were RBC count, hemoglobin concentration, hematocrit, MCV, MCH, MCHC, RDW, reticulocyte count, platelet count, differential WBC counts. The clinical pathology endpoints included in clinical chemistry were AST, CK, bilirubin, and creatinine in all studies and ALT and BUN in the Jackson Laboratory study.

We have added further details in the Materials and Method section lines 984-989 to address the above reviewer#4's requests for further precision and improve upon the clarity of the manuscript.

Comment 6: A comparison to a greater variety of rapidly dividing primary cultures (ie GI epithelium, hematopoietic, skin cells) typical of oncology target tissues would give much greater confidence in the safety of PCLX-100.

While the structure of the GI was analyzed by our pathologist and co-author Dr. Wei-feng Dong from the department of Oncology, we conceptually agree with reviewer#4 on this but the description of the impact of PCLX-001 on the numerous rapidly growing tissues from a toxicology perspective clearly belongs in a subsequent manuscript. Importantly, these studies are best performed under GLP by a certified laboratory and an extensive array of toxicological studies have recently been performed. We are awaiting the results and final report from the CRO. Some of the key results are also described and published in an Abstract form (see reference 55).

Comment 7: In fact, PBMCs (normal cells) and BL2 cell IC50's appear similar (~0.03-0.1 μ M) based on Figs 1 and S5. This should be stated in the absence of additional cytotoxicity assays in more normal cell types. Based on the benefit/risk for this therapeutic indication, it is acceptable and not unusual that some normal tissues are effected at efficacious doses.

We had already commented on the fact there is about a 50% of PBMCs that die in the 0.050- to 0.1 μ M PCLX-001 range but that the remaining cells are extremely resistant to PCLX-001 at concentrations up to 10 μ M in the results section lines 183-189.

At the request of the reviewer, we inserted the following sentence in lines 548-551 of the discussion:

“ In the absence of additional cytotoxicity assays in more normal cell types, based on the benefit/risk for this therapeutic indication, it is acceptable and not unusual that some normal tissues (e.g. blood cells including PBMCs) are effected at efficacious doses.”

Minor Comments:

Comment 8: The authors should be commended for conducting an extensive screen of kinases. Cross species pharmacology is not confirmed with this platform containing human kinases, and it should be described as a “human” kinase screen. Was the screening panel used in this study the preconfigured Eurofins “scanMAX” KINOMEScan or a custom panel of 468 kinases?

We agree with reviewer#4, our work represents a thorough analysis of the potential use of PCLX-001 as a first-in-kind NMT inhibitor in cancer starting with lymphoma.

We have now specified the type of assay performed by Eurofins as a “preconfigured scanMAX KINOMEScan containing 468 human kinases” on lines 225-227.

Comment 9: The authors use the term “robotic” repeatedly throughout the manuscript (12 times). It is not necessary to repeat this detail throughout the manuscript.

We have removed the superfluous “robotic” were applicable throughout the manuscript. We left three. One in the abstract, one in the introduction and one in the result sections.

Comment 10: Decreases in the expression in non-myristoylated proteins may be due to cell death, which often causes a general decrease in gene/protein expression (although increases in some specific proteins may also associated with cell death depending on mechanism/compensatory pathways). In fact, it appears that these decreases coincide with cytotoxic concentrations of PCLX-100. The authors should discuss/address this possibility.

At the request of the reviewer, we have now discussed this possibility in lines 560-577:

“While PCLX-001 treated cells still remained at least 75% viable at concentrations of PCLX-001 that are starting to become cytotoxic, whether the lower downstream signaling protein levels correspond to a reduction in gene transcription or increased protein degradation in dying cells is not known.”

Comment 11: What was the rationale for using SQ route of administration for xenograft studies? The text states that PCLX-100 is an orally bioavailable small molecule, which suggests clinical dosing will be by the oral route?

When the original experiments were performed we did not know whether or not PCLX-001 would be equally bioavailable whether delivered subcutaneously or orally. Now we know that it is. This will be reported in a subsequent manuscript on AML to be submitted in the fall 2020.

Comment 12: (Lines 145-149) Why use 100 nM vs 0.1uM? I suggest using consistent units throughout.

We have corrected all the nM throughout when referring to PCLX-001 concentration.

Comment 13: Resistant PBMCs tolerated up to 10 uM (Lines 149-151). What about resistant tumor cells (ie BL2)? Why point out this observation without a comparison to resistant tumor cells?

At 10 μ M for 4 day treatment, all BL2 cells are dead (figure 1E) while this treatment only kills about half the population of PBMCs. There are therefore no such BL2 cells resistant to PCLX-001 at 10 μ M. This comment is made to highlight the apparently large therapeutic window of PCLX-001 in vitro, which hopefully will translate *in vivo* in upcoming human clinical trials.

Comment 14: (Lines 290-293) Comparison's to dasatinib and Ibrutinib are really comparing potency. Due to tolerability considerations, it may be possible that these other kinase inhibitors are better tolerated and can be dosed at higher levels than PCLX-100 leading to more efficacy. The important consideration is efficacy compared to toxicity. The authors should describe that this is at the same concentrations and not make general statements about more effectively killing tumor cells in general.

We agree. To accommodate the reviewer's comments we made the following changes:

"Effectively" was removed from line 402 at the recommendation of the reviewer.

The sentence "Altogether, PCLX-001 has the broadest spectrum of efficacy against malignant lymphoma cell lines at both 48 and 96hrs in comparison to dasatinib and ibrutinib"

was also changed for:

"Altogether, PCLX-001 has the broadest spectrum of potency against malignant lymphoma cell lines at both 48 and 96hrs in comparison to dasatinib and ibrutinib" in lines 415-417.

Comment 15: (Line 340) – "in" can be removed before apoptosis (may be an orphan from previous revisions)

Agreed, "in" was removed in line 370 now.

Comment 16: (Line 372-376) – Correct for accuracy PBMC's (normal cells) similar in sensitivity, whereas IM9 are not "normal" cells, but an immortalized cell line.

The reviewer is correct, for accuracy and succinctness the original sentences "For the first time, we demonstrate that cancer cells can be selectively killed by a NMT inhibitor at concentrations lower than that required to kill normal cells. Indeed, the dose of PCLX-001 inducing lymphoma cell cytotoxicity and inhibiting proliferation was substantially lower in malignant cells compared to benign, immortalized B-cell controls (Fig. 1E-H, S3, S4)."

were changed to:

Lines 545-548: “We demonstrate that cancer cells can be selectively killed by a NMT inhibitor at concentrations lower than that required to kill and inhibit the proliferation of immortalized and normal cells (Fig. 1E-H, S3, S4).”

Comment 17: (Line 594) – Should be binding vs “biding”

Edited accordingly on now line 852

Comment 18: (Line 733) – Why did a clinical pathologist review the slides versus an anatomical pathologist?

We thought a trained clinical pathologist in the department of Oncology would be superior at discerning the possible effects (significant or subtle) caused by our compound on tissues. We simply thought someone used to look at tissues from patients treated with various chemotherapeutic agents such as Dr. Wei-feng Dong would be best for this task.